# NANOG is repurposed after implantation to repress *Sox2* and begin pluripotency extinction

Frederick C K Wong[1,2,8], Man Zhang[2,3,8], Ella Thomson [ID][2,4], Linus J Schumacher[4,5], Anestis Tsakiridis[2,6], James Ashmore[2,4], Tong Li[1], Guillaume Blin [ID][2,4], Eleni Karagianni[2,7], Nicholas P Mullin[2,4], Ian Chambers [ID][4✉] & Valerie Wilson [ID][2,4✉]

## Abstract

**Loss of pluripotency is an essential step in post-implantation development that facilitates the emergence of somatic cell identities essential for gastrulation. Before implantation, pluripotent cell identity is governed by a gene regulatory network that includes the key transcription factors SOX2 and NANOG. However, it is unclear how the pluripotency gene regulatory network is dissolved to enable lineage restriction. Here, we show that SOX2 is required for post-implantation pluripotent identity in the mouse, and cells that lose SOX2 expression in the posterior epiblast are no longer pluripotent. Using in vitro and in vivo analyses, we demonstrate anticorrelated expression of NANOG and SOX2 preceding gastrulation, culminating in an early disappearance of pluripotent identity from posterior NANOG[high]/SOX2[low] epiblast. Surprisingly, *Sox2* expression is repressed by NANOG and embryos with post-implantation deletion of *Nanog* maintain posterior SOX2 expression. Our results demonstrate that the distinctive features of post-implantation pluripotency are underpinned by altered functionality of pluripotency transcription factors, ensuring correct spatio-temporal loss of embryonic pluripotency.**

**Keywords** SOX2; NANOG; Embryo; Pluripotency; Post Implantation
**Subject Categories** Chromatin, Transcription & Genomics; Development

## Introduction

Pluripotency is the ability of a cell to give rise to differentiated progeny of all three embryonic germ layers and is a specific attribute of epiblast cells of the early mammalian embryo (Smith, 2001). Cells of the pre- and early post-implantation epiblast have distinct pluripotent states referred to as naive and primed, respectively (Hackett and Surani, 2014; Nichols and Smith, 2009). These states exhibit transcriptional, epigenetic and functional characteristics that are mirrored by cell lines derived by ex vivo culture: embryonic stem cells (ESCs) from pre-implantation epiblast (Evans and Kaufman, 1981; Martin, 1981) and epiblast stem cells (EpiSCs) from post-implantation epiblast (Brons et al, 2007; Tesar et al, 2007). In the embryo, pluripotent cells can be detected until the onset of somitogenesis by their capacity to generate teratocarcinomas upon transplantation (Damjanov et al, 1971) and to form EpiSC lines in culture (Osorno et al, 2012).

A pluripotency gene regulatory network (PGRN) centred on three core transcription factors (TFs) OCT4, SOX2 and NANOG promotes pluripotency in ESC (Chambers and Tomlinson, 2009; Jaenisch and Young, 2008; Ng and Surani, 2011). Although many PGRN components are present in both naive and primed pluripotent states, distinct ESC- and EpiSC-specific TFs are also expressed (Brons et al, 2007; Festuccia et al, 2012; Greber et al, 2010; Guo et al, 2009; Tesar et al, 2007). While studies in ESCs have identified positive regulatory relationships between OCT4, SOX2 and NANOG, how post-implantation cells exit the pluripotent state, and whether pluripotency TF reconfiguration plays a role in this process is unclear. Intriguingly, however, spatially restricted expression patterns of these core components have been reported (Hoffman et al, 2013; Osorno et al, 2012; Sumi et al, 2013; Sun et al, 2014; Wood and Episkopou, 1999). Here, we provide evidence that the initial loss of pluripotency in the proximal posterior epiblast is caused by repression of *Sox2* by NANOG.

## Results

### NANOG and SOX2 expression are segregated in post-implantation embryos

Previous reports on expression of the core components of the PGRN have shown that NANOG and SOX2 are reciprocally expressed in gastrulating embryos (Chazaud et al, 2006; Mulas et al, 2018; Nichols et al, 2009; Plusa et al, 2008; Yamanaka et al, 2010). To determine whether TF reconfiguration plays a role in the loss of pluripotency, we examined the temporal dynamics of pluripotency TF expression in individual nuclei leading up to gastrulation.

[1]The Wellcome Sanger Institute, Wellcome Genome Campus, Cambridge CB10 1SA, UK. [2]Institute for Stem Cell Research, School of Biological Sciences, The University of Edinburgh, Edinburgh, UK. [3]Guangzhou National Laboratory, No. 9 XingDaoHuanBei Road, 510005 Guangzhou, China. [4]Institute for Regeneration and Repair, University of Edinburgh, 5 Little France Drive, Edinburgh EH16 4UU, UK. [5]School of Mathematics and Maxwell Institute for Mathematical Sciences, University of Edinburgh, Edinburgh, UK. [6]Centre for Stem Cell Biology, School of Biosciences, The University of Sheffield, Alfred Denny Building, Western Bank, Sheffield S10 2TN, UK. [7]Novo Nordisk Research Centre, Oxford, UK. [8]These authors contributed equally: Frederick C K Wong, Man Zhang. ✉E-mail: ichambers@ed.ac.uk; v.wilson@ed.ac.uk

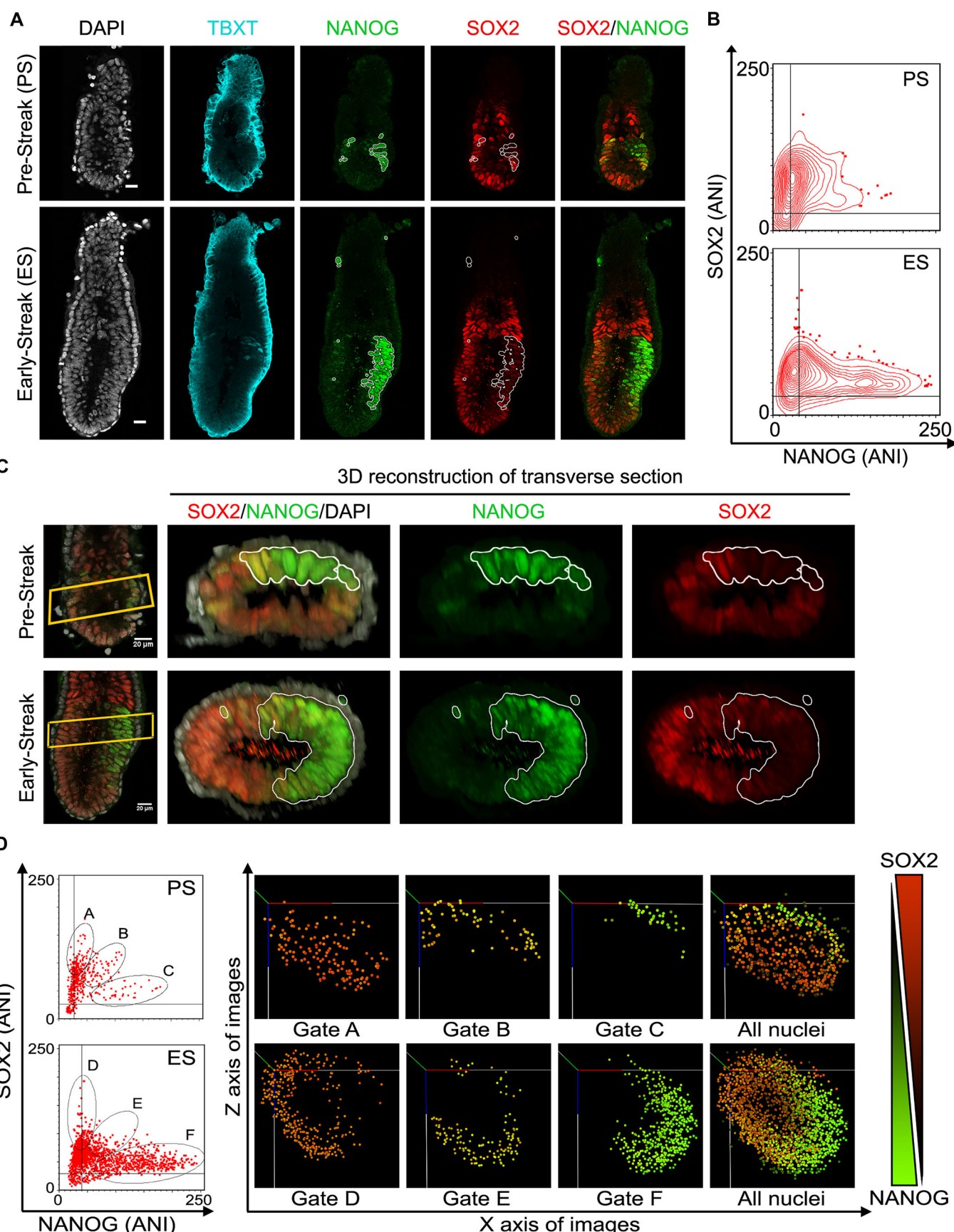

**Figure 1. SOX2 and NANOG proteins are negatively correlated before gastrulation begins.**

(A) Sagittal confocal section of pre-streak (T⁻ epiblast) and early-streak (T⁺ epiblast) ~E6.5 embryos stained for TBXT (cyan), NANOG (green), SOX2 (red) and DAPI (grey). NANOG-positive cells are outlined and overlaid on Sox2 channel: bar, 10 µm. (B) Quantitation of SOX2 and NANOG immunofluorescence for each stage (n = 2 embryos). Contours, 5% equal probability contouring; dots: outliers beyond lowest contour line. (C) A proximal region of the whole embryo confocal image (yellow box) was used to generate a 3D reconstructed transverse view. NANOG (green)-positive cells were outlined in white and overlaid onto SOX2 (red). The x axis of the 3D reconstructed views were aligned along the long axis of the embryos to show the transition of the Nanog⁺ posterior epiblast from the long axis observed in pre-streak-stage embryos to the short axis in early-streak-stage embryos. White scale bar = 20 µm. (D) Left, quantified immunofluorescence (ANI) of SOX2 and NANOG in pre-streak and early-streak embryos were gated for Sox2^HI (gates A, D), Sox2^MED/Nanog^MED (gates B, E), Nanog^HI (gates C, F) cells. Right, these gates were mapped onto the ANIs visualised in 3D space (axes in 3D space = X, Y, Z axes of the whole embryo confocal images) to show the spatial localisation of gated ANIs in the embryo. Orange-green colour gradient in the 3D space visualisation represents NANOG or SOX2 ANI level (high NANOG = green, high SOX2 = orange). Source data are available online for this figure.

Initially, a time course combining whole-mount immunofluorescence with quantification of fluorescence intensity was performed (Figs. 1A,B, EV1 and EV2). This showed the expected pre-implantation patterns of expression with NANOG and SOX2 protein levels positively correlating at E4.5 (Fig. EV1A–C). Following implantation, OCT4 and SOX2 were widely expressed in the epiblast (Fig. EV1A,B), but NANOG protein was undetectable at E5.5 (Fig. EV1A,B). The first cells in the post-implantation epiblast to express NANOG were in pre-streak embryos (E6.0). Interestingly, NANOG expression began before T (brachyury) (TBXT) expression (Fig. 1A,B) in nascent mesoderm of the primitive streak, marking the start of gastrulation (Fig. 1C,D). Even at the pre-streak stage, SOX2 protein levels were lower in cells with high NANOG, within 12 h of the onset of NANOG expression (Fig. 1A,B (PS)). At early-streak stage (E6.5), NANOG was maximally expressed in proximal posterior, SOX2^low epiblast cells prior to their entry into the primitive streak, (Figs. 1A–D (ES) and EV1A–C). By the neural plate stage (E7.5), the NANOG and SOX2 domains were clearly segregated, although OCT4 remained widely expressed (Fig. EV1A–C). Subsequently, by the head-fold stage, NANOG expression had diminished and the SOX2 domain extended posteriorly (Fig. EV1A–C). These results indicate that expression of NANOG and SOX2 proteins is already segregated before gastrulation, with the two proteins progressively occupying distinct territories in the developing post-implantation epiblast.

## The segregation of NANOG and SOX2 expression can be captured in epiblast stem cells

The changes in expression of SOX2 and NANOG in pre- and post-implantation embryos prompted a comparison of SOX2 and NANOG expression in corresponding established pluripotent cell lines (Fig. EV3A). Levels of SOX2 and NANOG mRNA and protein were high, and protein levels were positively correlated in individual ESC nuclei, whereas in EpiSCs, NANOG and SOX2 levels were reduced and no longer positively correlated (Fig. EV3A–C). To determine how these changes occurred during the naive-to-primed transition, core PGRN mRNA and protein were monitored following transfer from ESC to EpiSC medium (Fig. EV3D–F). Transcript levels of NANOG declined within 24 h, followed by SOX2, while Pou5f1 (and its protein product OCT4) decreased only slightly (Fig. EV3D). Known ESC-specific SOX2 target genes (Nakatake et al, 2006; Nishimoto et al, 1999; Shi et al, 2006; Tokuzawa et al, 2003) were not uniformly altered (mRNAs for Zfp42/Rex1 and Fbxo15 were downregulated, but Utf1 and Lefty1 were not; Fig. EV3D), suggesting distinct SOX2

dependencies in ESC and EpiSC. In contrast to the situation in ESCs, a negative correlation between NANOG and SOX2 became apparent on Nanog re-expression after 48–72 h in EpiSC medium (EV3E-F). These dynamic alterations in PGRN components in vivo and in vitro imply that regulatory interactions between core PGRN components are fundamentally altered during the naive-to-primed pluripotency transition.

## The SOX2^low posterior epiblast precociously loses pluripotency

At the neural plate stage of development (E7.0), SOX2 was virtually undetectable in the NANOG⁺ proximal posterior epiblast (region 1) (Fig. 2A,B), while OCT4 expression remained uniformly high (Fig. EV4A). Region 1 was microdissected from the rest of the epiblast (region R) for functional analysis (Fig. 2B). To confirm dissection accuracy and profile region 1 expression, we compared region 1 with region R by qRT-PCR. Region 1 expressed lower Sox2, higher Nanog, and equivalent Pou5f1 mRNA compared to region R (Fig. 2B). There was no enrichment of Prdm1, indicating no contamination of region 1 by PGCs (Fig. EV4B). Region 1 was also enriched for primitive streak marker Fgf4 and depleted for other markers of epiblast pluripotency and early neural identity (Sox3, Utf1 Fig. EV4C) while other Sox transcripts (Sox4, Sox11) were equally expressed in regions 1 and R. TBXT, as well as SOX3, which acts redundantly with SOX2 in primed pluripotent cells (Corsinotti et al, 2017), also showed reduced protein levels in region 1 (Fig. EV4D) (Acloque et al, 2011; Uchikawa et al, 2011; Wood and Episkopou, 1999).

Dissected regions were assessed for pluripotent function by teratocarcinoma-forming assays and by their ability to generate EpiSC lines upon explantation in vitro. Consistent with a previous study (Beddington, 1983), fragments of region 1 transplanted under the kidney capsule did not form tumours, in contrast to size-matched fragments of region R (Fig. 2C; see (Osorno et al, 2012) for teratocarcinoma scoring criteria). During ex vivo culture, a transient spike in expression of the Nanog-GFP reporter at 24–48 h of ex vivo culture predicts the ability of cells to establish pluripotent EpiSC lines, thereby providing a rapid in vitro test of prospective pluripotency (Osorno et al, 2012). While region R explants of Nanog-GFP embryos expressed GFP at 24 h, explants from region 1 did not (Fig. 2D). Importantly, selective labelling of the epiblast (by injection of DiI into the amniotic cavity) showed that Nanog-GFP was activated specifically in the explanted epiblast of region R, but not region 1 (Fig. EV4E), excluding the possibility that the absence of GFP reporter expression in region 1 was because

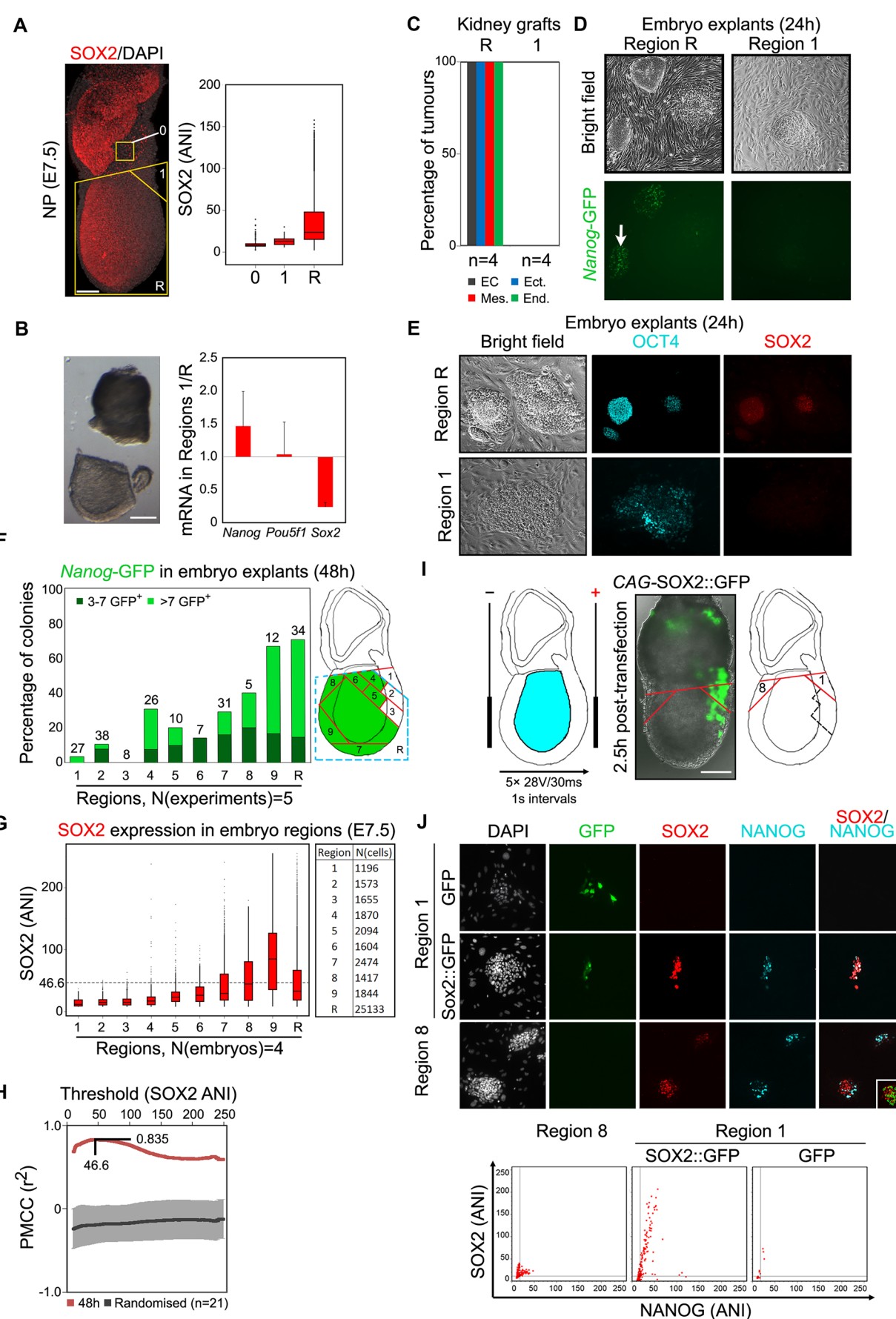

**Figure 2. SOX2 is required for post-implantation epiblast pluripotency.**

(A) SOX2 immunofluorescence (left), quantified (right) against a background extraembryonic region (0); grey = DAPI ($n = 3$ embryos). White scale bar = 50 μm. Boxplots: Box, 25–75% interquartile range (IQR); whiskers, remaining 5–95%; dots, outliers; line, median. (B) Quantitative mRNA analysis in dissected regions 1 and R. For *Sox2*, $n = 3$; *Oct4*, *Nanog*, $n = 2$. Error bars represent standard deviation; only positive value is shown for clarity. White scale bar = 50 μm. (C) Germ layer representation in teratoma assay of dissected regions as indicated. White arrow, colony reactivating >7 Nanog-GFP cells (see (F)). (D) Representative Nanog-GFP E7.5 (neural plate stage) explant cultures (24 h); a colony with >7 GFP-expressing cells is indicated. (E) Immunofluorescence of 24 h explants showing OCT4 (cyan) and SOX2 (red). (F) Proportion of GFP-expressing colonies after 48 h explant culture (total colony numbers analysed are indicated above each bar). Right: diagram of epiblast showing regions that reactivate *Nanog*-GFP (green) at 48 h. Light green = percentage of colonies with >7 GFP-expressing cells. Dark green = percentage of colonies with 3–7 GFP-expressing cells. (G) SOX2 ANI distribution ($n = 4$ embryos) across the corresponding dissected regions in the whole-mount embryo confocal stacks, number of cells analysed are indicated. Boxplots: as (A). (H) Pearson's correlation coefficient (PMCC, $r^2$) was calculated between the observed percentages of GFP+ colonies across all regional explants after 48 h (F) and the percentages of cells above a variable threshold SOX2 ANI level ((G), *y* axis) within their corresponding regions in the embryo (red line). A maximum $r^2$ of +0.835 was observed between the percentages of GFP+ colonies and of cells above 46.6 Sox2 ANI amongst the ten analysed embryonic regions (from $n = 3$ stained E7.5 embryos). A randomised control was performed to calculate the PMCC between the percentages of GFP+ colonies and of cells above the threshold of SOX2 ANI, but in a randomised order of regions. The mean average of this randomised PMCC across all threshold levels is plotted (dark grey line) along with the corresponding standard deviation (light grey error bars) representing several iterations of randomisation ($n = 21$). (I) E7.5 embryo electroporation by posterior-orientated anode of pPyCAG-SOX2-2A-GFP (SOX2::GFP) or pPyCAG-GFP control plasmids injected into the amniotic cavity (cyan). Proximal posterior (1) and proximal anterior (8) regions dissected for explant culture are indicated. Scale bar = 50 μm. (J) Explants imaged at 24 h for SOX2 (red) and NANOG (cyan) immunofluorescence and GFP fluorescence (green); explanted regions and transfected plasmids are indicated. Lower: SOX2 and NANOG ANI distributions in explanted colonies. Source data are available online for this figure.

epiblast cells were missing from these explants. Moreover, region R explants formed densely packed colonies expressing OCT4 and SOX2, whereas explants from region 1 lacked SOX2, expressed OCT4 only in a small subset of cells and had a dispersed colony morphology (Fig. 2E).

To determine the extent of the non-pluripotent region, ex vivo culture of epiblast regions of similar size to region 1 was performed (Fig. 2F). The number of GFP-positive cells per colony was assessed at 48 h (Figs. 2F and EV4F). Several regions produced a high proportion of GFP-positive colonies (Fig. 2F) that could be maintained for multiple passages as EpiSC lines expressing OCT4, SOX2 and NANOG (Fig. EV4G). Notably, however, posterior regions 1–3 had the lowest fraction of GFP-positive colonies at 48 h (Fig. 2F). These colonies lost the characteristic EpiSC morphology when passaged (Fig. EV4H). Regions 1–3 also had the lowest SOX2 protein level per nucleus (Fig. 2G). Activation of the GFP reporter was correlated with a threshold level of ~20% of maximum SOX2 immunofluorescence (Fig. 2H). Together, these results suggest that SOX2 is required for epiblast pluripotency, and that EpiSC formation from explants (prefigured by *Nanog*-GFP expression) requires the explanted cells to express sufficient SOX2.

## Enforced SOX2, but not OCT4, expression restores pluripotency to posterior epiblast

Previous studies have established that forced OCT4 expression can reverse the loss of pluripotency occurring at the onset of somitogenesis (Osorno et al, 2012). As OCT4 expression was diminished in region 1 explants (Fig. 2E), the pluripotency of the gastrulation-stage proximal posterior epiblast was reassessed following OCT4 induction. Embryos carrying a doxycycline-inducible *Pou5f1* transgene and a *Nanog*-GFP reporter (Hochedlinger et al, 2005; Osorno et al, 2012) (Fig. EV5A) were separated into posterior region 1–3 and remainder region for ex vivo culture (Fig. EV5B). As expected, region R produced more GFP-positive cells than regions 1–3 in uninduced explants (Fig. EV5C,D). However, OCT4 induction did not stimulate *Nanog*-GFP expression in explants from region 1 (Fig. EV5C,D). Therefore, enforced OCT4 expression cannot rescue pluripotency in explanted SOX2-

negative posterior epiblast. Furthermore, OCT4 induction in later embryonic tissues showing differential SOX2 expression resulted in Nanog reactivation only in SOX2-expressing epithelial tissues (Fig. EV5E–G). These results suggest that the establishment of ex vivo cultures of pluripotent cells requires not only OCT4 but also SOX2 expression.

The ability of enforced SOX2 expression to induce NANOG expression in explants of the SOX2-negative posterior epiblast was next examined. Constitutive SOX2 (SOX2::GFP) or control (GFP only) expression plasmids were injected into the amniotic cavities of neural plate-stage embryos and electroporated into the posterior epiblast. Following incubation (2.5–4 h), embryos expressing GFP (Fig. 2I) were dissected, and proximal posterior (region 1) and anterior embryonic (region 8) were explanted (Fig. 2I,J). As expected, most colonies from anterior explants expressed both SOX2 and NANOG after 24 h (Fig. 2J, 11/18 colonies, $n = 5$ embryos). While posterior explants transfected with control GFP plasmid generally did not reactivate NANOG (Fig. 2J, EV4I; 1/16 NANOG+ colonies, $n = 5$ embryos), posterior explants transfected with SOX2::GFP restored NANOG expression (Fig. 2J, EV4I; 18/26 NANOG+ colonies, $n = 5$ embryos). Together, these results imply a requirement not only for OCT4 (Osorno et al, 2012) but also for SOXB1 proteins (Corsinotti et al, 2017) for primed pluripotency.

## A novel in vitro system modelling epiblast patterning reveals that NANOG represses SOX2 and suppresses pluripotency

In *Nanog*$^{−/−}$ ESCs, inducing NANOG expression did not affect SOX2 expression (Festuccia et al, 2012). Moreover, ESCs genetically engineered to express differing NANOG levels expressed similar SOX2 protein levels (Fig. EV6A). Thus, there is no evidence that NANOG regulates SOX2 levels in naive pluripotent cells. While NANOG and SOX2 were not well-correlated in established EpiSCs (Fig. EV3B), an inverse relationship between NANOG and SOX2 was apparent from day 2 of ESC differentiation in EpiSC culture media (Fig. EV3E), similar to that in the ≥E6.5 embryo (Fig. 1B). The near-homogeneous formation of early post-implantation epiblast-like cells (EpiLCs) by transfer of ESCs previously cultured

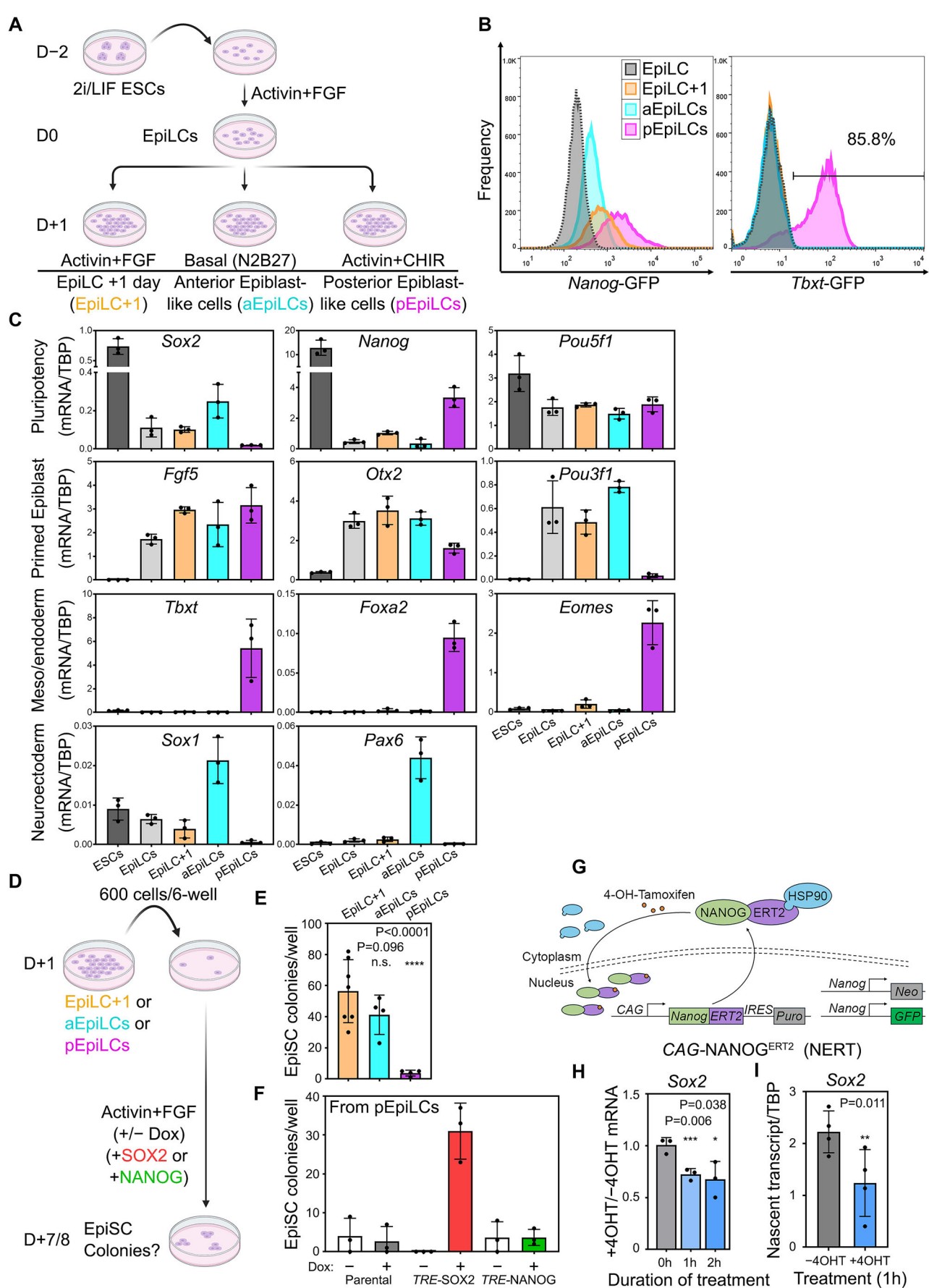

◄  **Figure 3.  In vitro differentiation mimics anteroposterior regionalisation of the post-implantation epiblast.**

(A) Experimental scheme showing EpiLC derivation from 2i/LIF cultured ESCs and subsequent differentiation to anterior epiblast-like cells (aEpiLCs), posterior epiblast-like cells (pEpiLCs) or a further culture in EpiLC condition for an extra day (EpiLC+1); EpiLCs are designated as day 0. (B) Flow cytometry analysis of *Nanog*-GFP expression in TNG cells or *Tbxt*-GFP-expressing cells in the indicated conditions at day +1, E14Tg2a cells was set as the negative gate. (C) Quantitative RT-PCR of day +1 E14Tg2a cells in the indicated culture conditions. (For all plots shown in C, E, F, H, I below: columns represent mean; error bars are standard deviation; points are individual biological replicates; $n \geq 3$). (D) Experimental scheme showing EpiSC colony formation assay from EpiLC+1, aEpiLCs or pEpiLCs starting populations. Cells were seeded at the indicated density, and the number of EpiSC colonies were scored after 6–7 days of culture in EpiSC medium with or without doxycycline-induction of transgene. (E) SOX2+/NANOG+/OCT4+ EpiSC colony numbers at day 7 from indicated starter populations cultured without doxycycline. (F) SOX2+/NANOG+/OCT4+ EpiSC colony numbers at day 8 after replating pEpiLCs into EpiSC condition. Doxycycline-mediated induction of SOX2 (*TRE*-SOX2) or NANOG (*TRE*-NANOG) is indicated. (G) The tamoxifen (4-OH-Tamoxifen)-inducible NANOG system. EpiLCs carrying *Nanog*$^{-/-}$; *CAG*$^{NanogERT2-IRES-puro}$ (NERT) are used to constitutively express NANOG-ERT2 fusion protein that is sequestered by HSP90 in the cytoplasm. The presence of tamoxifen releases NANOG-ERT2 from HSP90 resulting in nuclear translocation. (H, I) Quantitative RT-PCR of *Sox2* total mRNA (H) or nascent transcript (I) expression in NERT EpiLCs treated with pEpiLC medium containing (+4OHT) or without (−4OHT) tamoxifen for the indicated hours. Expression levels are normalised to TBP (I) and the −4OHT control (H). Two-sided unpaired *T* test significance ($P < 0.05$) is indicated by asterisks with the *P* value shown. Source data are available online for this figure.

in 2i/LIF into Activin/FGF (Hayashi et al, 2011) provides a robust starting point to investigate the early postimplantation induction of NANOG. We therefore assessed the consequences of manipulating EpiLC culture conditions on Nanog and Sox2 expression (Fig. 3A). Culturing EpiLCs for 24 h in EpiSC conditions (Activin and Fgf2) resulted in low-level expression of *Sox2* and *Nanog* (Figs. 3A–C and EV6C), hereafter termed EpiLC + 1 day (EpiLC+1). In contrast, 24 h in the presence of the Wnt/ß-catenin signalling agonist CHIR99021 and Activin, which mimics the signalling environment occurring in the Nanog-positive proximal posterior epiblast (Figs. 1C and 3A), efficiently upregulated *Nanog* as well as markers of mesoderm and endoderm (Figs. 3B,C and EV6B,C). Notably, *Sox2* mRNA was low (Fig. 3C) and SOX2 was absent from most cells, appearing prominently in cells lacking NANOG and T (Fig. EV6C). These features are reminiscent of posterior epiblast (posterior epiblast-like cells; pEpiLC). Conversely, 24-h culture of EpiLCs in N2B27 basal media induced neural markers *Sox1* and *Pax6*, mimicking anterior epiblast (anterior epiblast-like cells; aEpiLC) (Fig. 3A,C). In these conditions, *Sox2* was upregulated and *Nanog* was suppressed. Importantly, *Pou5f1* and *Fgf5* expression were maintained in all three conditions, indicating that epiblast identity was preserved in the population (Fig. 3C).

To determine whether anterior or posterior epiblast-like cells could form EpiSCs, we replated aEpiLCs, pEpiLCs or control EpiLC +1 at clonal density in EpiSC conditions for 6 days (Figs. 3D and EV7A). As expected, EpiLC+1 yielded Sox2+/Nanog+/Oct4+ EpiSC colonies (Figs. 3E and EV7B,C). A similar efficiency of EpiSC colony formation was found in aEpiLCs. However, colony formation was negligible in replated pEpiLCs, mimicking the early disappearance of pluripotency in the posterior epiblast in vivo (Figs. 3E and EV7B,C). Importantly, doxycycline-mediated induction of SOX2 (but not NANOG) in pEpiLCs rescued EpiSC colony formation (Figs. 3F and EV7D,E). These results indicate that without SOX2, pEpiLCs cannot transit to an EpiSC phenotype.

To assess the effect of NANOG on SOX2 and post-implantation pluripotency using our EpiLC-based system, we first determined the transcriptional response to NANOG using tamoxifen-mediated relocation of a NANOG-ERT fusion protein into the nucleus (Festuccia et al, 2012) in *Nanog*$^{-/-}$ pEpiLCs (Fig. 3G). This rapidly downregulated total *Sox2* mRNA to ~70% of EpiLC levels ($P < 0.05$) within an hour, suggesting that NANOG directly represses *Sox2* in pEpiLCs (Fig. 3H). This conclusion was reinforced by a reduction of nascent *Sox2* mRNA to 50% of EpiLC levels by 1 h (Fig. 3I).

Interestingly, although mRNA levels of the pluripotency/early neural markers *Sox3*, *Otx2*, and *Pou3f1* were not significantly altered 2 h following NANOG induction (Fig. EV7G), their nascent transcripts were significantly reduced within 1 h (Fig. EV7H). These data suggest that *Nanog* represses *Sox2* and other pluripotency/early neural markers, likely at the transcriptional level.

Bulk RNA-seq comparison of wild-type aEpiLC with pEpiLC transcriptomes showed that additional known markers of anterior and posterior epiblast were upregulated in the expected conditions (Fig. 4A; Dataset EV1). Moreover, GO terms associated with mesoderm and endoderm formation were enriched in pEpiLCs, whereas neural development-associated terms were enriched in aEpiLCs (Fig. 4B,C). This reinforced the conclusion that aEpiLC and pEpiLC conditions accurately model anterior and posterior epiblast formation from EpiLCs.

We next examined the dynamic relationship between NANOG and SOX2 proteins in a time course of pEpiLC differentiation (Fig. 4D,E). NANOG was upregulated within 3 h of transfer to pEpiLC conditions. By 6 h, NANOG-SOX2 anticorrelation emerged (Fig. 4E, arrows). This pre-dated the induction of mesoderm, evidenced by the onset of TBXT expression between 6 and 24 h (Fig. 4E, insets), and a near-complete segregation of NANOG-high from SOX2-expressing cells (Fig. 4E, contour plot). To test whether the emerging NANOG-SOX2 anticorrelation in pEpiLC conditions marked a change towards a posterior epiblast cell state, we mined a mouse gastrulation-stage single-cell RNA-seq atlas ((Pijuan-Sala et al, 2019), Dataset EV2). Genes whose expression was significantly correlated with *Nanog* and anticorrelated with *Sox2* in the epiblast between E6.5-7.5 at ≥2 timepoints were considered as candidate co-regulated genes (here termed *Nanog*-correlated genes). These included components of NODAL/TGFB and FGF signalling (*Tdgf1, Fst, Fgf8*), and genes involved in mesoderm or endoderm differentiation (e.g. *Eomes, Fgf8*). Conversely, genes anticorrelated with NANOG and correlated with SOX2 at ≥2 timepoints (*Mycn, Utf1, Pou3f1*, and *Dnmt3b*) were associated with pluripotency and/or anterior neural differentiation (*Sox2*-correlated genes; Figs. 4F,G and EV8A). To validate these *Nanog*- and *Sox2*-correlated genes, we determined their localisation in a separate spatial transcriptomic dataset (Wang et al, 2023). *Sox2*- or *Nanog*-correlated gene expression colocalised with either *Sox2* or *Nanog*, respectively, in agreement with the single-cell RNA-seq correlations (Fig. EV8B,C). Like *Sox2* and *Nanog* themselves, *Sox2*-

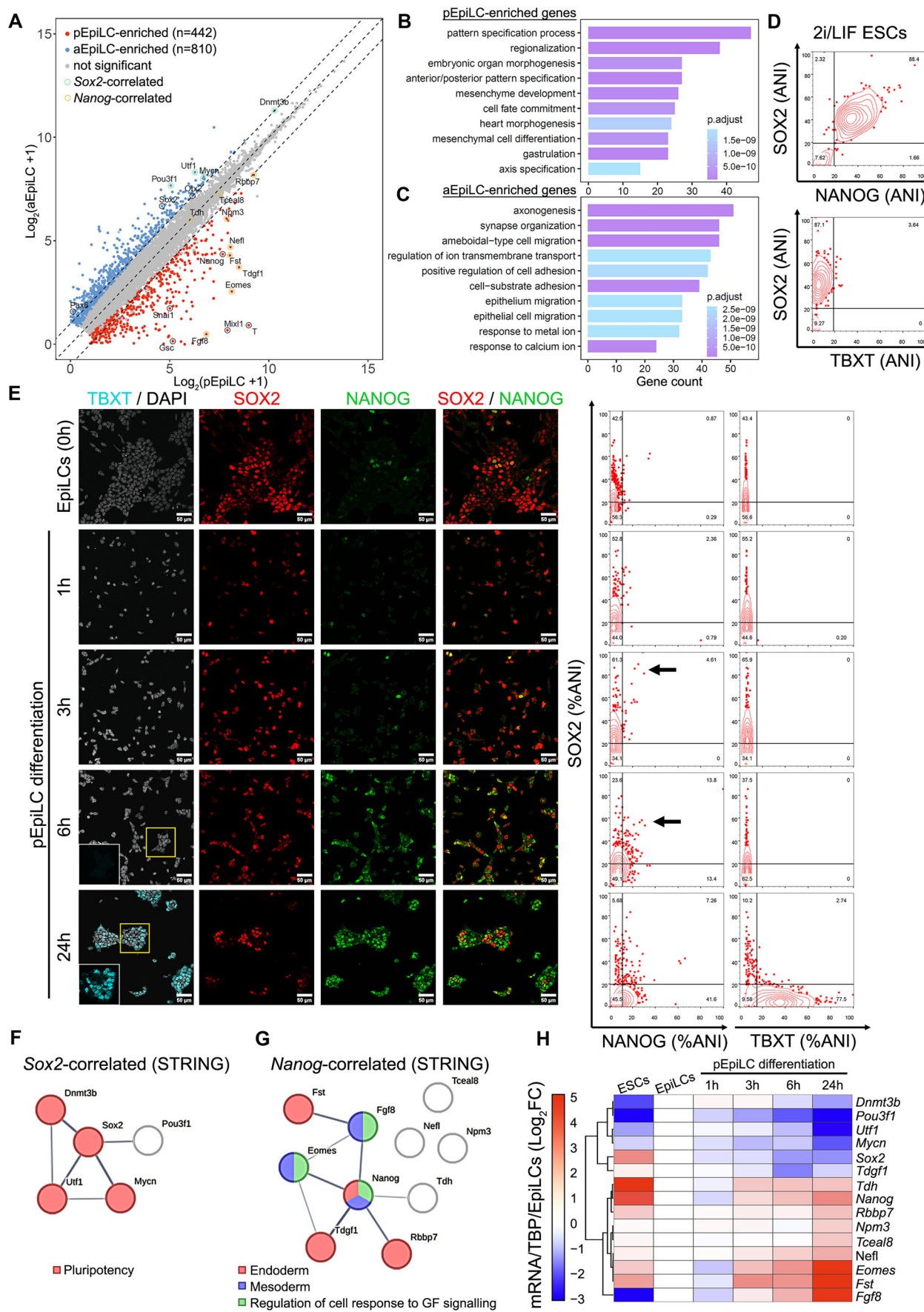

◄ **Figure 4. In vitro pEpiLC differentiation recapitulates aspects of gastrulation.**

(A) Differentially expressed genes between pEpiLC (x axis) and aEpiLC (y axis). Significant enrichment (*P* < 0.05) is highlighted in blue (aEpiLC) or red (pEpiLC). Genes of interest are labelled. (B, C) GO enrichment in pEpiLC (B) and aEpiLC (C). (D) 10% contour plots of SOX2, NANOG and TBXT expression (ANI) in ESCs cultured in 2i+LIF prior to time-course analysis, showing all events. (E) Time course immunofluorescence of EpiLC cultured in pEpiLC condition for 24 h. Cells were fixed and stained at the indicated hours for TBXT (cyan), SOX2 (red), NANOG (green) and DAPI (grey). SOX2/NANOG/TBXT (%ANI) are plotted on the right as 10% contour plots showing all events. Arrows at 3 and 6 h: highest SOX2 expression levels in NANOG-positive cells. Scale bars: 50 μm. (F) STRING network of known interactions between *Sox2* and *Sox2*-correlated genes. The total confidence score of interactions between each node is represented by thin (≥0.4) or thick (≥0.7) edges. Common enriched GO terms shared between ≥3 nodes are depicted by common node colours. GO terms are indicated. (G) STRING network between *Nanog* and *Nanog*-correlated genes. (H) Heatmap showing qRT-PCR measurement of *Nanog*- and *Sox2*- correlated gene expression over the pEpiLC differentiation time course. Colours represent positive (red) or negative (blue) fold change versus EpiLCs. Profiles at *t* = 1–24 h are hierarchically clustered using cosine similarity. See Fig. EV9D for individual gene expression plots. Source data are available online for this figure.

correlated and Nanog-correlated genes were elevated in anterior and posterior epiblast, respectively, before overt mesoderm formation (≤E6.5; Fig. EV8B,C). Bulk RNA-seq analysis of aEpiLC and pEpiLC transcriptomes (Fig. 4A) additionally showed that *Nanog*- and *Sox2*-correlated genes were upregulated in pEpiLC and aEpiLC conditions, respectively.

We assessed the changes in *Nanog*- and *Sox2*-correlated gene expression by qRT-PCR of cells during the pEpiLC differentiation time course. Sox2-correlated genes were downregulated from 3 h onwards (Figs. 4H and EV8D), when NANOG protein level increases (Fig. 4E) consistent with the idea that NANOG represses a network of pluripotency/early neural-associated genes (Trott and Martinez Arias, 2013; Vallier et al, 2009). Interestingly, 5/9 of the *Nanog*-correlated genes (*Eomes, Fst, Nefl, Rbbp7* and *Tdh*) were, like Nanog, upregulated at 3 h after transfer to pEpiLC conditions. As *Nanog*-correlated genes overlap extensively with genes previously shown to be upregulated by ectopic Nanog expression in vivo between E4.5 and E7.5 ((Tiana et al, 2022); Fig. EV8E; *Eomes, Fst, Nefl, Tdgf1, Fgf8 and Npm3*), these observations suggest that our *Nanog*-correlated gene list is enriched for targets of NANOG. *Sox2*-correlated genes may also include negatively regulated targets of NANOG. Of these, only *Pou3f1* is also downregulated upon Nanog induction in vivo (Tiana et al, 2022). Taken together with the alterations in nascent transcripts 1 h following NANOG induction (Figs. 3H,I and EV7G,H), this suggests NANOG regulates genes involved in epiblast patterning. For instance, NANOG may directly repress *Pou3f1*, and indirectly induce *Eomes*. Nevertheless, we identify a primary role for *Sox2* as a direct NANOG target.

To investigate potential effects of *Nanog*-correlated genes on *Sox2* in the epiblast in vivo, we employed partial information decomposition (PID) analysis (Williams and Beer, 2010), which quantifies non-linear correlations and has been used to infer gene regulatory networks. This showed that *Nanog*, together with the *Nanog*-correlated genes, both synergistically and redundantly predicted *Sox2* expression better than *Tbxt*, whose expression appeared later than the *Nanog-Sox2* anticorrelation but increased with time (Fig. EV8F). Collectively, these observations suggest that suppression of SOX2 by NANOG may be enhanced by other posterior epiblast-specific factors, including components of the NODAL and FGF pathways.

### NANOG represses SOX2 expression in vivo

To test whether NANOG suppresses SOX2 in vivo, we electroporated NANOG-GFP and control GFP-expressing constructs into the

anterior, SOX2-high, epiblast at E7.25 (Fig. 5A). After a 6 h culture period, we performed immunofluorescence for NANOG, GFP and SOX2 followed by image analysis (Figs. 5B and EV9A–C). This showed that anterior cells expressing ectopic NANOG-GFP expressed lower levels of SOX2 than their neighbours (*n* = 3 embryos; 460 NANOG-GFP+ cells), while cells expressing control GFP expressed similar levels of SOX2 to their neighbours (*n* = 2 embryos, 537 GFP+ cells) (Figs. 5B and EV9D,E). Therefore, NANOG suppresses SOX2 in the epiblast in vivo as well as in vitro, suggesting that the early disappearance of pluripotency in the posterior epiblast is at least in part due to NANOG repressing SOX2.

To assess the effect of *Nanog* removal upon SOX2 expression in vivo, mouse lines carrying a *lox*P-flanked *Nanog* allele and Sox2-Cre (Hayashi et al, 2002) were intercrossed (Fig. EV10A–C). Analysis of streak-stage (E7.0–7.5) embryos showed that controls carrying Sox2-Cre but retaining one or two wild-type *Nanog* alleles excluded SOX2 from the posterior epiblast as expected (Figs. 5C, EV10D, EV11B). In contrast, Sox2-Cre+ gastrulation-stage embryos where *Nanog* was deleted not only retained SOX2 in the posterior epiblast (Figs. 5C, EV10D, and 11A–D), but also showed ectopic SOX2 expression in mesoderm (Figs. 5C and EV11A,B). Only a few cells escaped *Nanog* deletion and these all expressed low SOX2 (Fig. EV11C,D). *Nanog*-deleted embryos, although slightly delayed relative to wild-type littermates (Fig. EV12A,B), showed correct morphological anteroposterior pattern, TBXT expression, and expression of Nanog-GFP from the recombined allele (Fig. EV10D,E). Thus, the posterior localisation of SOX2 does not result from a failure to form an anteroposterior axis. In summary, Nanog excludes Sox2 from the posterior epiblast and consequently, in contrast to its pre-implantation activity in preserving pluripotency, Nanog plays a key role in dismantling the restructured post-implantation pluripotency network during gastrulation.

## Discussion

SOX2 and NANOG are core members of the pre-implantation pluripotency network. Our results show that after implantation, NANOG suppresses SOX2 to negatively regulate pluripotency. The naive and primed states feature different cytokine dependence (Tesar et al, 2007), enhancer usage (Tesar et al, 2007; Yeom et al, 1996) and SoxB1-level dependence (Corsinotti et al, 2017). This study shows that these changes are underpinned by an altered PGRN TF architecture that includes a shift in the spectrum of NANOG and SOX2-correlated genes compared to ESCs (Festuccia et al, 2012). NANOG- and SOX2- correlated genes do not overlap with genes that are regulated by OCT4 during gastrulation,

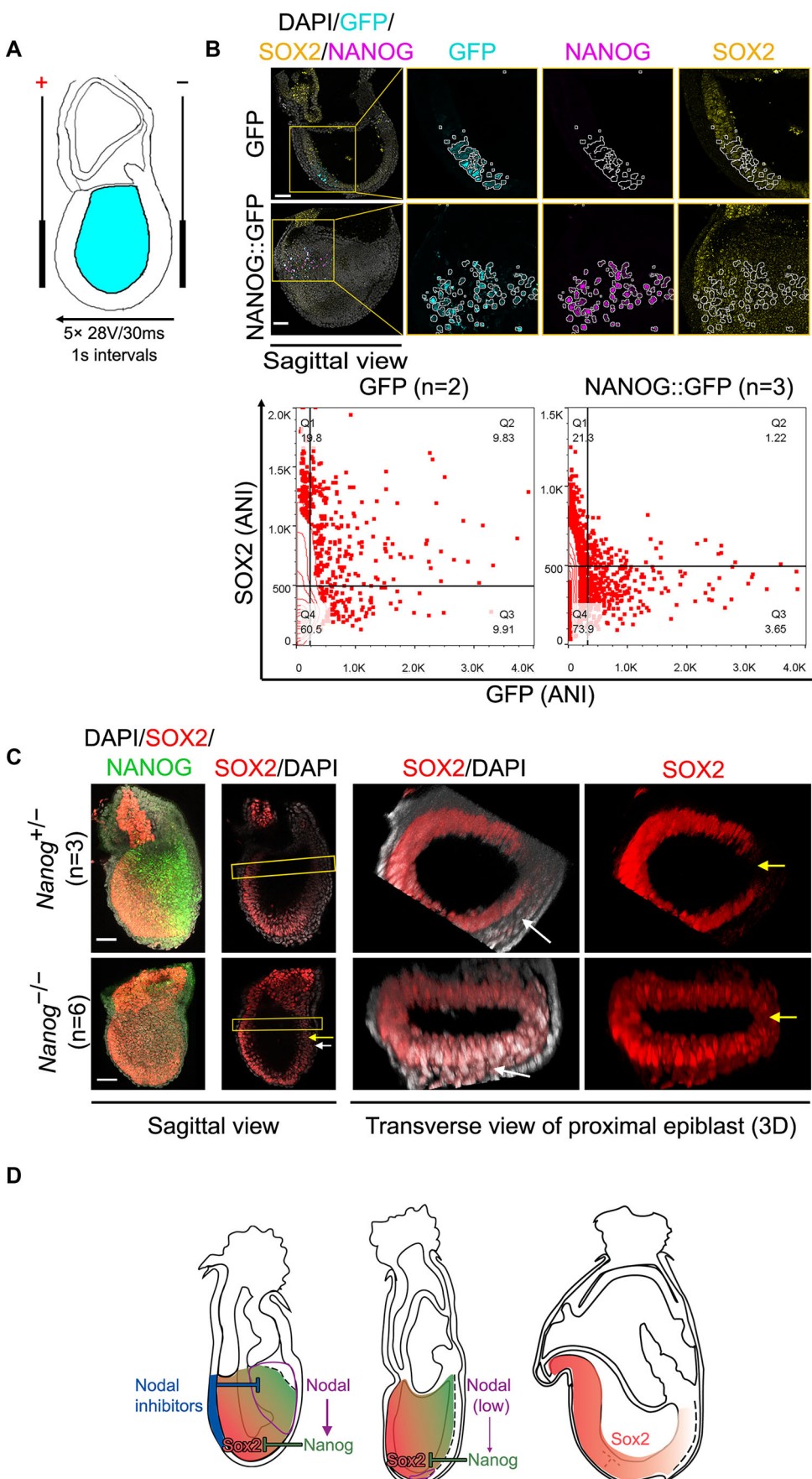

**Figure 5. NANOG negatively regulates SOX2 expression in post-implantation pluripotent cells.**

(A) E7.5 embryo electroporation by anterior-orientated anode of pPyCAG-NANOG-2A-GFP (NANOG::GFP) or pPyCAG-GFP control plasmids injected into the amniotic cavity (cyan) in a manner similar to Fig. 2I. (B) A Z-slice of a representative embryo from each electroporation is shown. Expression of GFP (cyan), exogenous NANOG (magenta) and SOX2 (yellow) are shown in the electroporated region (orange box inset) with GFP+ cells outlined in white to show the presence/absence of NANOG and/ or SOX2. White scale bar = 50 μm. DAPI-stained nuclei was segmented and quantified for SOX2 and GFP ANI (n = 2 for *CAG*-GFP embryos; n = 3 for *CAG*-NANOG::GFP embryos, see Fig. EV10), and their relationship is plotted with a 10% contour plot showing all events. (C) Left, confocal immunofluorescence of NANOG (green) and SOX2 (red) in E7.5 *lox*P-flanked *Nanog/*Sox2-Cre embryos (n = 3 for Nanog⁺/⁻ embryos; n = 6 for Nanog⁻/⁻ embryos). Scale bar = 50 μm. Right, 3D rendered transverse sections (orange box) show SOX2 in the posterior epiblast (yellow arrow) and mesoderm (white arrow). (D) Model showing the roles of SOX2 and NANOG in gastrulating embryos (details in text). Source data are available online for this figure.

identified in a previous study (Mulas et al, 2018). Thus, although in primed pluripotent cells, expression of SOX2 and NANOG is reduced compared to naive cells, primed pluripotency does not simply represent a weakened naive PGRN. The peri-implantation "formative" pluripotent stage, equivalent to EpiLCs (Smith, 2017), is characterised by minimal NANOG levels. This suggests that continuous changes in the PGRN prepare epiblast cells to exit the naive pluripotent state, with extinction of pluripotency then being facilitated by re-expression of NANOG in a re-wired PGRN context.

Figure 5D integrates our findings with published in vivo data. Between E5.5-E6.0 NANOG is re-expressed in response to Nodal/ Activin activity that restricts NANOG expression posteriorly (Camus et al, 2006; Osorno et al, 2012; Perea-Gomez et al, 2002; Sun et al, 2014; Vallier et al, 2009). Pre-gastrulation, posterior SOX2 diminishes in NANOG-positive cells and is emphatically eliminated during gastrulation (Fig. 1). Together with the demonstration that the epiblast loses pluripotency upon NANOG expression, and regains pluripotency upon SOX2 expression, this argues that NANOG exerts its effects on pluripotency not by diverting cells to mesoderm or endoderm differentiation but by inhibiting SOX2. The effect of NANOG on SOX2 transcription is rapid, leading to a 50% reduction in nascent SOX2 transcript, and over the next 24 h, SOX2 protein is reduced to near-background levels. NANOG may also negatively regulate a set of genes, either directly or indirectly, that includes *Sox3, Pou3f1, Otx2* and *Utf1* which initially maintain epiblast pluripotency, but later form part of an early neural programme. The protein product of *Pou3f1*, OCT6, positively regulates *Sox2* (Iwafuchi-Doi et al, 2012; Matsuda et al, 2017), suggesting that NANOG could downregulate SOX2 via suppression of OCT6. Our data shows that NANOG reduces *Pou3f1* transcript concurrently with *Sox2*, supporting a direct repression of SOX2 by NANOG, rather than an indirect one via OCT6 downregulation. However a later OCT6 effect that reinforces this initial downregulation cannot be ruled out. Interestingly, Mulas et al (Mulas et al, 2018) showed that loss of OCT4 function in the epiblast disrupted A–P axis orientation without abrogating SOX2-NANOG anticorrelation, indicating that OCT4 is not an essential mediator of SOX2 repression by NANOG.

A previous study (Sumi et al, 2013) showed that WNT pathway stimulation in early-gastrulation E6.5 embryos led to rapid, ubiquitous NANOG expression and complete repression of SOX2, while WNT pathway repression resulted in near-ubiquitous SOX2 and diminished NANOG expression. This suggests that WNT signalling independently positively regulates NANOG. This would be consistent with the correctly localised expression of TBXT in *Nanog*⁻/⁻ embryos in this study. However, the suppression of SOX2 by NANOG in the low-WNT

environment of the anterior epiblast argues that NANOG mediates SOX2 suppression independently of WNT. Segregation of *Nanog*-expressing from *Sox2*-expressing cells is also seen in (Trott and Martinez Arias, 2013). Interestingly, in these experiments, the addition of the WNT pathway agonist CHIR99021 cannot maximally induce *Tbxt* nor suppress *Sox1*, in *Nanog*⁻/⁻ cells. Therefore, NANOG may facilitate optimal WNT responsiveness to prepare cells for an early lineage choice between a posterior fate (towards primitive streak differentiation) and an anterior fate (towards neural differentiation). The expression of SOX2 in the primitive streak and nascent mesoderm in *Nanog*⁻/⁻ embryos thus suggests that NANOG is required as part of the circuitry to ensure SOX2 shutdown as cells pass through the primitive streak.

During gastrulation, SOX2 and NANOG inhibit the differentiation of mesodermal and neural lineages, respectively (Vallier et al, 2009; Wang et al, 2012), while NANOG supports definitive endoderm differentiation (Teo et al, 2011). Therefore, the emergence of segregated domains expressing NANOG (where mesoderm and endoderm-fated cells originate) and SOX2 (corresponding to a region of neural fate) (Lawson et al, 1991) may initiate the earliest lineage decisions of the pluripotent post-implantation epiblast. Thus, the regulatory interactions described here provide key mechanistic insight into regionalised lineage restriction at gastrulation.

# Methods

**Reagents and tools table**

| Reagent/resource | Reference or source | Identifier or catalogue number |
|---|---|---|
| **Experimental models** | | |
| E14TG2a mESCs | Hooper et al, 1987 | tg2a mESCs |
| Nanog-ERT2 mESCs | Festuccia et al, 2012 | ESΔN-NERT mESCs |
| MF1 wild-type mice | | MF1 |
| 129/Ola wild-type mice | | 129/Ola |
| Nanog:GFP mice | Osorno et al, 2012 | TNG |
| Sox2:Cre mice | Hayashi et al, 2002 | Sox2:Cre |
| Nanog flox/flox mice | Chambers et al, 2007 | Nanog flox/flox |
| Sox2:Cre; NanogΔ/ + mice | This study | Sox2:Cre; NanogΔ/+ |
| tgOct4 mice | Osorno et al, 2012 | tgOct4 |

| Reagent/resource | Reference or source | Identifier or catalogue number |
|---|---|---|
| **Recombinant DNA** | | |
| Sox2-T2A-GFP plasmid | This study | |
| GFP-T2A-GFP plasmid | This study | |
| Nanog-T2A-GFP plamid | This study | |
| **Antibodies** | | |
| Oct4 antibody | Santa Cruz | N-19 |
| NANOG antibody | eBiosciences | 14-5761 |
| SOX2 antibody | Abcam | ab92494 |
| SOX2 antibody | Abcam | ab97959 |
| GFP antibody | Abcam | ab38689 |
| T/Brachyury antibody | R&D | AF2085 |
| SOX3 antibody | Abcam | ab42471 |
| β-Actin antibody | Abcam | ab20272 |
| **Software** | | |
| ImageJ/FIJI | NIH | https://fiji.sc/ |
| Prism | GraphPad | https://www.graphpad.com/ |
| Photoshop | Adobe | https://www.adobe.com |
| Cellpose | Stringer et al, 2021 | http://cellpose.org |
| FlowJo | BD | http://flowjo.com |
| Farsight Toolkit | Rey-Villamizar et al, 2014 | https://github.com/RoysamLab/Farsight-toolkit |
| **Chemicals, Enzymes** | | |
| ActivinA | Peprotech | 120-14-10UG |
| Chiron 99021 | Stratech | B5779-APE |
| bFGF | Lifetech | 13256029 |
| Tamoxifen | Sigma-Aldrich | H7904-5MG |
| **Other** | | |
| SP8 Confocal Microscope | Leica | |
| LightCycler 480 Instrument II | Roche | |
| Odyssey Fc | LI-COR | |
| Fortessa Flow Cytometer | BD Biosciences | |
| FACSAria II Cell Sorter | BD Biosciences | |
| **Oligonucleotides** | | |
| Please refer to Dataset EV3 | | |

## Reagents and resources

### ESC culture

ESCs were cultured on gelatin-coated dishes in 100U/ml LIF/10% FCS/β-mercaptoethanol/GMEM (LIF/FCS) unless otherwise stated.

For ESC culture adaptation to LIF/BMP or LIF/2i, media was changed to LIF (100U/ml)/BMP4 (100 ng/ml)/N2B27 (Ying et al, 2003) or LIF (100U/ml)/1 µM PD0325901/3 µM CHIR99021/N2B27 (Ying et al, 2008), respectively, and ESCs cultured for >3 passages before use.

### ESC differentiation

For differentiation into EpiSCs, ESCs were replated on gelatin-coated dishes at $3.1 \times 10^3$ cells/cm$^2$ and media changed to ActivinA (20 ng/ml)/ bFGF (10 ng/ml)/N2B27 after 24 h. Cells were either analysed every 24 h or collected with Accutase (Sigma) and passaged as clumps onto fibronectin-coated dishes after 4 days. Cells for analysis are lysed directly for mRNA quantification, fixed and stained for immunofluorescence imaging, or replated in LIF/FCS at 62 cells/cm$^2$ for 8 days before being stained for alkaline phosphatase activity (Sigma 86R-1KT). Cells were initially passaged 1:20 every 2 to 3 days. Stably propagating EpiSCs were maintained by passaging 1:10 every 2 to 3 days with media changes daily. ESC-derived EpiSCs were cultured on dishes pre-coated with 7.5 µg/ml fibronectin without feeders in ActivinA (20 ng/ml)/ bFGF (10 ng/ml)/N2B27 (Osorno et al, 2012) for >9 passages.

To derive EpiSCs de novo a region of E7.5 embryonic tissue was dissociated into clumps prior explanting and expanding on feeders in KOSR-containing Activin/FGF medium as described (Osorno et al, 2012). Stably propagating EpiSCs on feeders were adapted to feeder-free culture by first changing media to Activin/FGF/N2B27 for 24 h before passaging onto fibronectin-coated plates with 1:5 density of feeders for 2–3 days, followed by passaging onto fibronectin-coated plates without feeders.

Tamoxifen-induced nuclear localisation of Nanog was achieved by transiting the previously described ESΔN-NERT ESCs (Festuccia et al, 2012) to EpiLCs. Briefly, ESΔN-NERT ESCs were converted to EpiLCs by plating $2.5 \times 10^4$ cells/cm$^2$ onto fibronectin-coated dishes in ActivinA (20 ng/ml)/ bFGF (12 ng/ml)/KSR (1%)/N2B27 for 48 h (Hayashi et al, 2012). To induce nuclear localisation of Nanog-ERT2, EpiLCs were collected and $5 \times 10^4$ cells/cm$^2$ replated onto fibronectin-coated dishes in EpiLC medium. After attachment (1 h), cells were treated with 4-hydroxy-tamoxifen (4OHT) at a final concentration of 1 µM for the indicated times prior to cell lysis and RNA preparation.

For pEpiLCs and aEpiLCs differentiation, day 2 EpiLCs were cultured in KSR (1%)/N2B27 medium plus ActivinA (20 ng/ml)/CHIR (3 µM/ml) and KSR (1%)/N2B27 medium respectively. For EpiSCs colony assay, 600 cells were plated in each well of six well plate within EpiSCs medium and fixed with 4% PFA after 6 (wild-type cells) or 7 days (KH2 doxycycline-inducible cell lines) for immunostaining.

## Embryo manipulation

All mouse studies were performed under UK Home Office project licence PPL 60/4435. *M. musculus* domesticus wild-type strains were either MF1 or 129/Ola. Nanog:GFP mice were derived using TNG ESCs as described (Chambers et al, 2007; Osorno et al, 2012). Ubiquitously induced Oct4 expression was achieved using TgOct4 homozygous males crossed with either 129/Ola or Nanog:GFP as described (Osorno et al, 2012). To delete *Nanog* post-implantation, Sox2:Cre mice (Hayashi et al, 2002) were crossed with *Nanog*flox/flox mice (Chambers et al, 2007) to obtain Sox2:Cre; *Nanog*Δ/+ males.

Mice were maintained on a 12 h light/dark cycle. E0.5 is designated as noon on the day of vaginal plugs. Embryos were dissected using standard methods (Nagy et al, 2003). Embryos were staged according to the morphological landmarks described in (Lawson and Wilson, 2025). ES, early streak; MS, mid streak; LSEB, late streak early bud transition; NP, neural plate; LNP, late neural plate; EHF, early head-fold; LHF, late head-fold. Collected E7.5 embryos were microdissected as shown in Fig. 2A,B using hand-pulled glass needles as described (Osorno et al, 2012).

To assess for regional pluripotency in NP stage E7.5 epiblasts, Nanog:GFP embryos were microdissected and explanted on feeders as described (Osorno et al, 2012). Triturated region R tissues were diluted 1:10 before explanting to account for the size difference. GFP fluorescence was imaged after 48 h, and images were analysed using Photoshop (Adobe). In brief, the boundary of each colony was outlined using the bright-field image. GFP+ foci in the corresponding GFP fluorescence image were identified using the Colour Range command, and the same settings were applied throughout the experiment. Identified foci outside the colony boundary were considered debris. Colonies with 1–2 foci were indistinguishable from debris and were excluded in this analysis. Double asterisks (**) denote $P$ value < 0.05 using the G-test goodness-of-fit approximation to the $\chi^2$ distribution.

To label NP stage E7.5 epiblasts prior to microdissection, DiI (Molecular Probes) was injected into the amniotic cavity using micropipettes as described (Wilson and Beddington, 1996). To prevent inadvertent contact between the DiI-loaded micropipette and embryonic endoderm/mesoderm, the micropipette was inserted via the extraembryonic region through the amnion into the amniotic cavity. Remaining unbound DiI was diluted below detectable levels upon microdissection in excess media.

To electroporate NP stage E7.5 embryos, plasmids expressing GFP or Sox2-T2A-GFP were prepared in PBS/0.01% Fast Green FCF dye (Sigma) at a final concentration of 1 µg/µl. DNA/dye mixtures were injected into the amniotic cavity as described for DiI injection. Embryos were placed between two parallel gold-tipped, 0.5 mm diameter electrodes (BTX Genetrode) set 5 mm apart with the posterior embryo facing the anode in PBS or M2 medium. Current was applied through in five pulses of 28 V for 30 ms each. Treated embryos were immediately transferred to static four-well dishes containing 50% rat serum and cultured in a humidified incubator at 5% $CO_2$. Only embryos that showed GFP expression in the posterior side after 2–4 h culture were used.

For germ layer separation, dissected forebrain and tail regions from E8.5 embryos were incubated in trypsin/pancreatin on ice for 20 min, followed by manual dissection with forceps and a hand-pulled glass needle to separate epithelia from the underlying mesenchyme. Tissues were explanted onto feeders as described above.

To delete Nanog post implantation, Sox2:Cre; $Nanog^{\Delta/+}$ males were mated with $Nanog^{flox/flox}$ females and Sox2:Cre; $Nanog^{\Delta/\Delta}$ post-implantation embryos collected at E7.5 for analysis. $Nanog^{flox/flox}$ mice were derived using the Nanog-conditional vector described (Chambers et al, 2007). Ear notches from juvenile mice or embryonic yolk sacs were incubated with lysis buffer containing 0.45% (v/v) Tween 20, 0.45% (v/v) NP40 and 0.5 mg/ml Proteinase K (Sigma, P2308) at 56 °C (2 h). Proteinase K was then heat-inactivated at 95 °C (10 min). After centrifugation, 5 µl of lysate was used directly for genotyping using the primers listed in Dataset

EV3. PCRs were performed with Taq Polymerase (Qiagen 201205) in a final volume of 30 µl. PCR products were separated on 2% TBE agarose gels and visualised via Ethidium Bromide staining.

## Teratocarcinoma induction

Kidney capsule grafts were performed as described (Osorno et al, 2012) (Tam, 1990) except that 129/Ola mice were used. Two pieces of region 1, one piece of region R or $5 \times 10^6$ EpiSCs were transferred per mouse. Mice were sacrificed 4 weeks post-engraftment. Histological staining on kidney sections was performed as described (Osorno et al, 2012). Morula aggregations (Nagy et al, 2003), blastocyst injections and embryo transfer were performed using standard procedures.

## Immunohistochemistry

Embryos and cells were stained as described (Osorno et al, 2012). Embryo sectioning and cryo-section staining were performed as described. DAPI-stained E3.5 to E5.5 embryos were mounted in PBS for imaging, whereas E6.5 to E7.75 embryos were incubated through a methanol/PBS series of 30%, 50%, 80% and twice in 100% methanol before mounting in a 1:2 (v/v) mix of benzyl-alcohol:benzyl-benzoate (BABB) clearing solution prior to imaging. Zona pellucidae of stained E3.5 blastocysts were removed with acidic Tyrode's solution (Sigma) immediately before imaging. Embryos were imaged using the TCS SP8 inverted confocal microscope (Leica) equipped with HCX IRAPO L ×25/0.95 W (E3.5 to E5.5 embryos) or HC PL APO ×20/0.70 IMM (E6.5 to E7.75 embryos) objectives. For high-resolution (hi-res) imaging of Nanog electroporated embryos, stained embryos were cleared using CUBIC (Susaki et al, 2015). Embryos were immobilised in a sagittal orientation between two rectangular glass coverslips (Brand BR470820) in a chamber made using two layers of 0.12-mm-thick SecureSeal imaging spacers (Sigma GBL654008) before imaged with a ×25 objective in overlapping (15%) tiles and merged. Anteroposterior (sagittal) positioning of cleared, DAPI-stained embryos on the coverslip can be visualised using a keychain UV LED light. Primary antibodies used on cells were: Sox2, 0.28 µg/mL (Abcam ab92494); Nanog, 1 µg/mL (eBioscience 14-5761); Oct4 (N-19), 0.5 µg/mL (Santa Cruz sc8628); βIII-tubulin, 0.67 µg/mL (Covant MMS-435P); GFP, 1 µg/mL (Abcam ab38689). Primary antibodies used on whole-mount embryos were: Sox2, 0.55 µg/mL (Abcam ab92494); Sox2, 5 µg/mL (Abcam ab97959), Nanog, 2.5 µg/mL (eBioscience 14-5761), Oct4 (N-19), 1 µg/mL (Santa Cruz sc8628), T/brachyury, 5 µg/mL (R&D AF2085) and Sox3, 2.5 µg/mL (Abcam ab42471). All pre-implantation embryos were stained with the monoclonal Sox2 (Abcam ab92494) antibody as the polyclonal Sox2 (Abcam ab97959) antibody can only be used in post-implantation embryos due to the presence of non-specific staining in blastocysts.

Photoshop (Adobe) was used to compile and process images.

## Immunofluorescence quantification

Collected stack images were processed as described (Osorno et al, 2012) with the following changes. Stacked 3D images were additionally corrected for scattering per channel using the following equation: corrected intensity = intensity/e$^{(-k \times depth)}$ and using the following constants (k): 0.001 (Alexa488), 0.009 (Alexa568), 0.002 (Alexa647) and 0.014 (DAPI). Incorrectly segmented nuclei were rectified using Nucleus Editor from the

Farsight toolkit (https://github.com/RoysamLab/Farsight-toolkit). To do this, 32-bit images of segmented nuclei were converted without scaling using Fiji ImageJ (http://fiji.sc/) to 16-bit for use with Nucleus Editor. Calculated average pixel intensities of 3D composite images were exported to Excel using an updated version of the Java application described previously. For high-res merged confocal stacks, cell segmentation utilised the CellPose (Pachitariu and Stringer, 2022) version 2.2.1, a deep learning framework with a pre-trained model (cyto2) for single-cell segmentation from the DAPI channel per confocal slice. Cell size was set at 35 pixels, with default parameters. After segmentation, features characterising cell morphology and protein expression were extracted using the 'scikit-image' Python package version 0.20.0 (van der Walt et al, 2014). Features included cell area, perimeter, eccentricity, and intensity distribution. Feature data in TSV format were then converted to flow cytometry FCS files using Text2FCS software (https://flowjo.typepad.com/the_daily_dongle/2012/10/new-build-of-text-2-fcs.html). Data were analysed using FlowJo (Tree Star) and represented in 5% or 10% contour plots showing outlier events. Quantified ANIs were gated using an internally controlled negative region per embryo (Figs. EV1D, 2, and 10B). In the analysis of $Nanog^{-/-}$ embryos, the number of detected Nanog-positive cells may be an underrepresentation due to segmentation sensitivity. Data were analysed using FlowJo (Tree Star) and represented in 5% or 10% contour plots showing outlier events. Quantified ANIs were gated using an internally controlled negative region per embryo (Figs. EV1D, 2, and 10B). In the analysis of $Nanog^{-/-}$ embryos, the number of detected Nanog-positive cells may be an underrepresentation due to segmentation sensitivity.

## FACS

Flow cytometry and cell sorting were performed using the Fortessa and Aria II (Becton Dickinson), respectively. Cells were resuspended in PBS/FCS (2% v/v) and stained with 0.1 µg/ml DAPI prior to FACS.

## Immunoblot analysis

Cell lysis and total protein preparation were performed as described (Gagliardi et al, 2013) except that 40 µg of total protein was analysed/lane on 10% Bolt SDS-PA gels (Novex) and transferred onto Whatman-Protran nitrocellulose membrane (Capitol Scientific). Antibody-stained membranes were analysed using the Odyssey Fc system (LI-COR Biosciences). Primary antibodies used were: Sox2, 0.11 µg/mL (Abcam ab92494); Nanog, 0.5 µg/mL (Bethyl A300-397A); β-actin, 50 ng/mL (Abcam ab20272).

## mRNA quantification by QRT-PCR

Cells were lysed directly on the culture plates and total RNA was prepared using the RNeasy mini kit (QIAGEN). 1 µg or 2 µg (cells) or 100 ng (isolated embryonic regions from pooled streak-stage embryos) total RNA per sample was reverse transcribed using SuperScript III (Invitrogen) and diluted 1:10 (cells) or 1:5 (embryonic regions). Each QRT-PCR reaction contained 3 µl diluted cDNA, 0.5 µM of each primer and LightCycler 480 SYBR Green I master mix (Roche). Reactions were performed and analysed in triplicate using LightCycler 480 II (Roche). Primers

used in this study are listed in Dataset EV3. All values plotted represent mean averages between the indicated number of experimental replications.

## Nascent RNA quantification

The Click-it nascent RNA capture kit (ThermoFisher Scientific, C10365) was used to measure newly synthesised RNA. In total, $5 \times 10^5$ ESΔN-NERT EpiLCs were seeded in N2B27 + 1%KSR with ActivinA (20 ng/ml) and Chiron (3 µM) medium for 15 min to allow cells to attach. Cells were then treated with 2.5 µl EU with or without 4-hydroxy-tamoxifen (4OHT) at a final concentration of 1 µM. Cells treated with 4OHT but without EU were used as negative controls. Cells were harvested after 1 h for RNA extraction. EU-labelled RNAs were purified and cDNAs synthesised according to the kit protocol. Four independent repeats of the experiment were carried out.

## Bioinformatics analysis

To find genes correlated with *Nanog* and *Sox2* expression in pEpiLC conditions, we used processed count data downloaded with the MouseGastrulationData package (Griffiths and Lun, 2024) from a study of mouse gastrulation and early organogenesis (Pijuan-Sala et al, 2019). We focused on cells denoted 'epiblast' from mouse embryos at stages E6.5-7.5. Using Spearman's rank correlation, we identified gene pairs whose expression was significantly correlated with either *Nanog* or *Sox2*. We used the correlatePairs function from the scran package (Lun et al, 2016) to calculate these correlations for cells at each embryonic stage, accounting for the mouse of origin. Genes were considered significantly correlated if their Benjamini-Hochberg corrected *P* value was below 0.05. To analyse gene lists of Nanog- or Sox2-correlated genes, we used the STRING web server package (Snel et al, 2000; Szklarczyk et al, 2023). Intersections between gene lists were determined using Venny 2.1 (Oliveros, 2007–2015). Heatmap was generated using the Morpheus web interface (https://software.broadinstitute.org/morpheus/). For bulk RNA-seq analysis, samples were sent to Majorbio for library construction and sequenced using $2 \times 150$ bp paired-end sequencing. The clean reads were mapped to the mouse genome (GRCm38) using STAR (V2.7.9a) (Dobin et al, 2013) Raw counts for each gene were estimated using RSEM (V1.3.1) (Li and Dewey, 2011) from the aligned BAM files. Differentially expressed gene analysis was performed using edgeR (V3.36.0) (Robinson et al, 2010).

## Partial information decomposition analysis

Partial information decomposition (PID) was quantified using the Julia package InformationMeasures.jl (https://github.com/Tchanders/InformationMeasures.jl) using the "Bayesian blocks" discretisation mode (Chan et al, 2017).

## Statistical analyses

Replicates and statistical tests are described in the figure legends. Graphing and statistical analyses were performed using Excel and GraphPad Prism. No sample size estimate or blinding was performed. Samples were only omitted from the analysis, retrospectively, if a clear confounding factor was identified.

## Data availability

RNA-sequencing data have been deposited at GEO with accession number GSE276569. Figure 2A, 5B,C source confocal data have been deposited at BioImage Archive with accession number S-BIAD2129 and can be accessed at https://doi.org/10.6019/S-BIAD2236.

The source data of this paper are collected in the following database record: biostudies:S-SCDT-10_1038-S44318-025-00527-9.

## Peer review information

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

## Acknowledgements

We thank Ron Wilkie for histology and Sally Lowell, Donal O'Carroll and Steve Pollard for comments on the manuscript. We acknowledge the help of staff from the imaging, FACS and Biomedical and Veterinary Services Facilities. This research was funded by grants from the Medical (VW and IC) and the Biotechnological and Biological Sciences Research Councils of the UK (IC) and by an MRC PhD Studentship and Centenary Award (FCKW).

## Author contributions

**Frederick C K Wong**: Conceptualisation; Data curation; Formal analysis; Validation; Investigation; Visualisation; Methodology; Writing—original draft; Writing—review and editing. **Man Zhang**: Conceptualisation; Resources; Data curation; Formal analysis; Validation; Investigation; Visualisation; Methodology; Writing—original draft; Writing—review and editing. **Ella Thomson**: Formal analysis; Investigation; Writing—review and editing. **Linus Schumacher**: Formal analysis; Visualisation; Methodology; Writing—review and editing. **Anestis Tsakiridis**: Formal analysis; Investigation; Writing—review and editing. **James Ashmore**: Software; Formal analysis; Methodology; Writing—review and editing. **Tong Li**: Software; Visualisation. **Guillaume Blin**: Resources; Software; Writing—review and editing. **Eleni Karagianni**: Methodology. **Nicholas P Mullin**: Investigation. **Ian Chambers**: Conceptualisation; Resources; Supervision; Funding acquisition; Writing—original draft; Project administration; Writing—review and editing. **Valerie Wilson**: Conceptualisation; Resources; Data curation; Formal analysis; Funding acquisition; Visualisation; Writing—original draft; Project administration; Writing—review and editing.

Source data underlying figure panels in this paper may have individual authorship assigned. Where available, figure panel/source data authorship is listed in the following database record: biostudies:S-SCDT-10_1038-S44318-025-00527-9.

## Disclosure and competing interests statement

The authors declare no competing interests.

# Expanded View Figures

**Figure EV1. Supporting Fig. 1.**

(A) Whole-mount embryo immunofluorescence showing OCT4 (cyan), SOX2 (red), NANOG (green) and DAPI (grey). Embryos were imaged by confocal microscopy and are shown as maximum Z-stack projections. Scale bars represent 20 μm. (B) Quantitation of SOX2/OCT4 and SOX2/NANOG immunofluorescence; number of embryos (brackets) and cells analysed (top right of righthand plot) are indicated. Gates were set based on the background fluorescence levels in the negative regions (see Fig. EV1D for example) per embryo. Quantified ANIs are represented in a 5% contour plot showing all events. (C) Pearson's correlation coefficients (PMCC, $r^2$) were calculated between SOX2 and NANOG ANI distributions at E4.5 ($n = 8$) and E6.5 ($n = 7$). (D) Regions cropped for whole-mount immunofluorescence quantification of E3.5–5.5 and E7.5-7.75. A separate internal negative region (0') was used for OCT4 ANI quantification at E3.5. Whole embryo images were quantified for E3.5 to E5.5 embryos, embryonic regions E were quantified for E6.5 to E7.75 embryos. A superficial z-slice was included to show lateral cells in E7.5 embryos (inset).

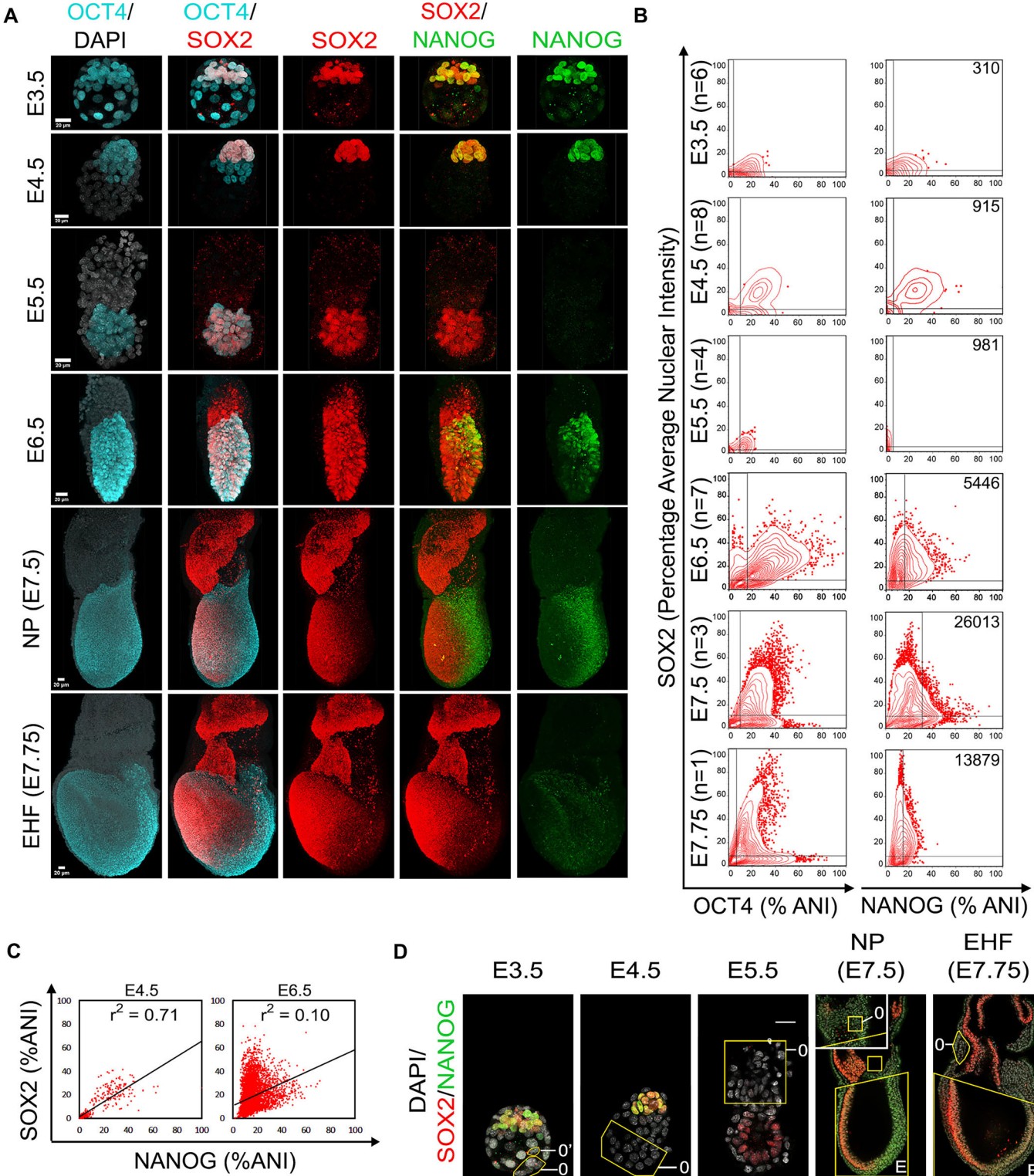

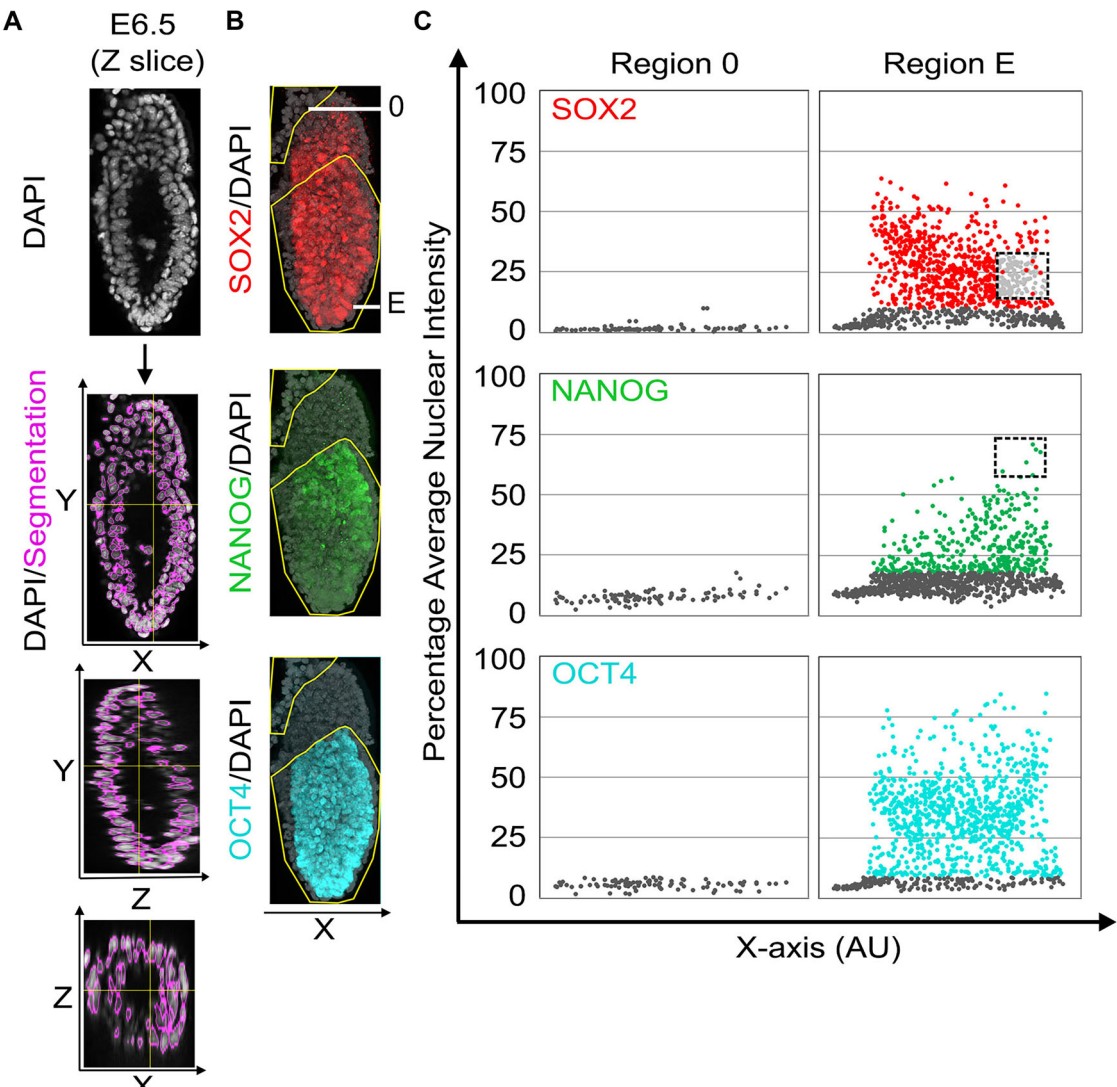

**Figure EV2. Supporting Fig. 1.**

(A) Workflow to quantify immunofluorescence per nucleus. The outline of each segment is shown (magenta). Orthogonal sections from a confocal Z-stack showing nucleus segmentation in three dimensions. (B) Regions cropped for whole-mount immunofluorescence quantification are shown by yellow outlines. An internal negative control region marked 0 was used to set gates for each embryo. (C) Average immunofluorescence signals from the three different channels shown in (B) were quantified by normalising to the nuclear volume to determine the average nuclear intensity (ANI) for each nucleus. ANIs were normalised to the maximal ANI per channel observed in all the embryos in the dataset shown in Fig. EV1A,B. ANIs from regions 0 and E are plotted along the x axis; the horizontal width of the corresponding images in (B). Nuclei in region E with ANI less than or equal to the maximum level observed in region 0 were considered negative (dark grey dots). Cells with the highest NANOG ANI (dashed black box in Nanog plot) in the proximal territory were assessed for SOX2, and the corresponding Sox2 ANI of these cells are shown (red inside dashed black box in SOX2 plot).

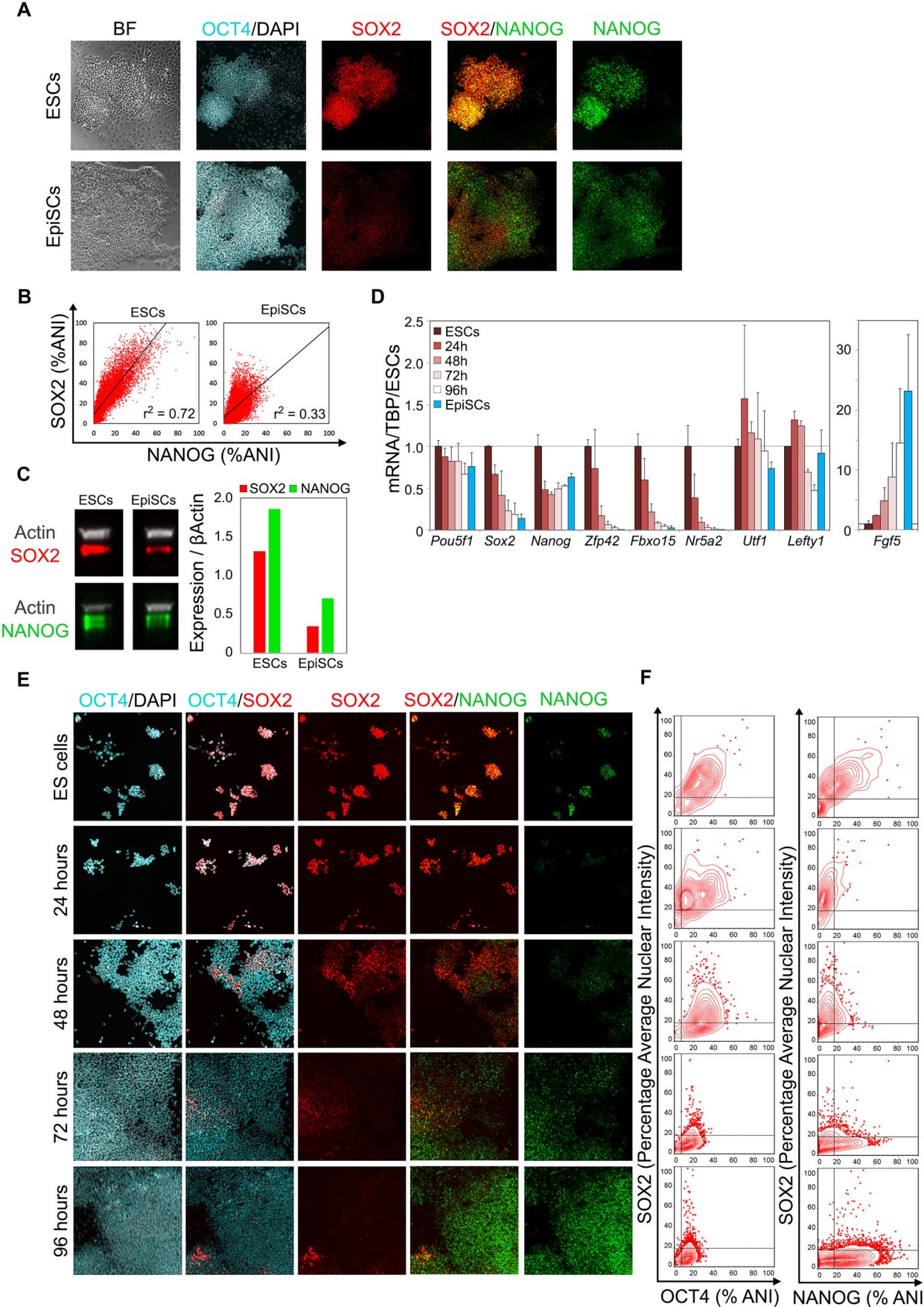

◀  **Figure EV3.  NANOG and SOX2 are partially segregated in EpiSCs.**

(A) Immunostaining for OCT4 (cyan), SOX2 (red) and NANOG (green) in E14TG2a ESCs and in EpiSCs. Nuclei were stained with DAPI (grey) (representative of $n = 26$ fields in 2 experimental replicates). (B) PMCC, ($r^2$) values were calculated between SOX2 and NANOG ANI distributions in ESCs ($n = 13011$ cells) and EpiSCs ($n = 9793$ cells analysed). (C) Total protein was extracted from ESCs and EpiSCs and analysed by immunoblotting for SOX2 (red), NANOG (green) and β-ACTIN (grey), lanes were cropped from the same blot. Right: quantification of SOX2 or NANOG band intensity relative to that of β-ACTIN. Shown is a representative of two independent replicate experiments. (D) Quantitative RT-PCR was performed for *Pou5f1*, *Sox2*, *Nanog*, *Zfp42 (Rex1)*, *Fbxo15*, *Nr5a2*, *Utf1*, *Lefty1* and *Fgf5* transcripts at the indicated times during the ESC to EpiSC transition. Transcript levels were normalised to TBP and plotted relative to ESC levels (black horizontal line). Error bars denote standard deviation (ESC, $n = 3$, 24-96 h, $n = 2$, EpiSC, $n = 4$ experimental replicates). (E) Immunostaining for OCT4 (cyan), SOX2 (red) and NANOG (green) during the ESC to EpiSC transition in E14TG2a cells. Nuclei were stained with DAPI (grey). (F) ANIs of SOX2/OCT4 and SOX2/NANOG were quantified for each 24 h time-point and represented in a 5% contour plot showing all events.

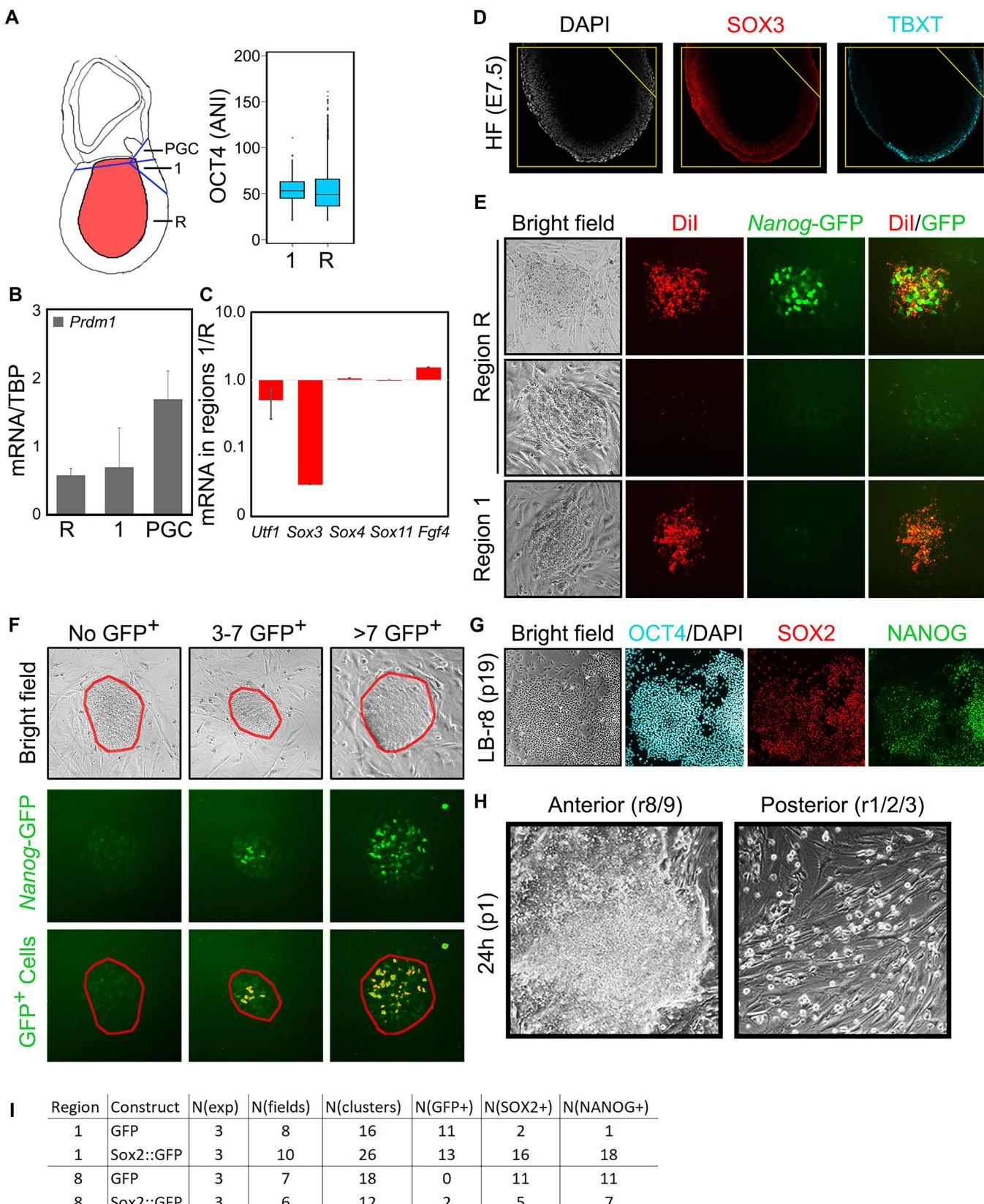

**Figure EV4. Supporting Fig. 2.**

(A) Left: Schematic diagram showing regions 1, R and 'PGC' for dissection. Red shading denotes DiI label injected into the amniotic cavity. Right: ANI distribution of OCT4 immunofluorescence in regions 1 and R of two representative embryos. Background ANI values determined using the internal negative control region 0 (Fig. EV1D) for each embryo were excluded in this plot. (B) QRT-PCR analysis of *Prdm1* (*Blimp1*) mRNA levels in regions 1, R and a PGC-containing region proximal to region 1 that includes the base of the allantois ($n = 2$). Error bars = SD. (C) Quantitative RT-PCR analysis of *Utf1, Sox3, Sox4, Sox11*, and *Fgf4* mRNA levels in regions 1 and R from dissected E7.5 embryos. For *Utf1*, $n = 2$; others, $n = 1$. Error bars = SD. (D) SOX3 (red) and TBXT (cyan) immunofluorescence at EHF stage. (E) *Nanog*-GFP was only observed in cells labelled with DiI after 24 h in culture (top row versus middle). Triturated region R tissue was diluted 1:10 prior to explantation; Region 1 was undiluted. (F) Colonies identified morphologically in bright field (red outline; top row) were examined for *Nanog*-GFP fluorescence (middle row). Using a constant threshold of green intensity, areas containing pixels with green intensity above this threshold were outlined (bottom row). The number of outlined areas in each outlined colony were scored and grouped into categories with no GFP+, 3–7 GFP+ or more than 7 GFP+ cells. Colonies with fewer than 3 GFP+ cells were indistinguishable from those containing only debris and were excluded from analysis. (G) Feeder-free EpiSCs (p19) were derived from expanding explants from one single region 8 of an LB-stage embryo (LBr8.3 cells). These cells were positive for OCT4 (cyan), SOX2 (red) and NANOG (green) immunofluorescence; (grey = DAPI). (H) The morphologies of representative colonies from the indicated regions 24 h after the first passage. (I) Quantification of explanted colonies (clusters) derived from indicated embryo regions expressing GFP, SOX2 and/or NANOG in ($n = 3$) electroporation experiments as depicted in Fig 2J. Numbers of total fields of view per condition are indicated.

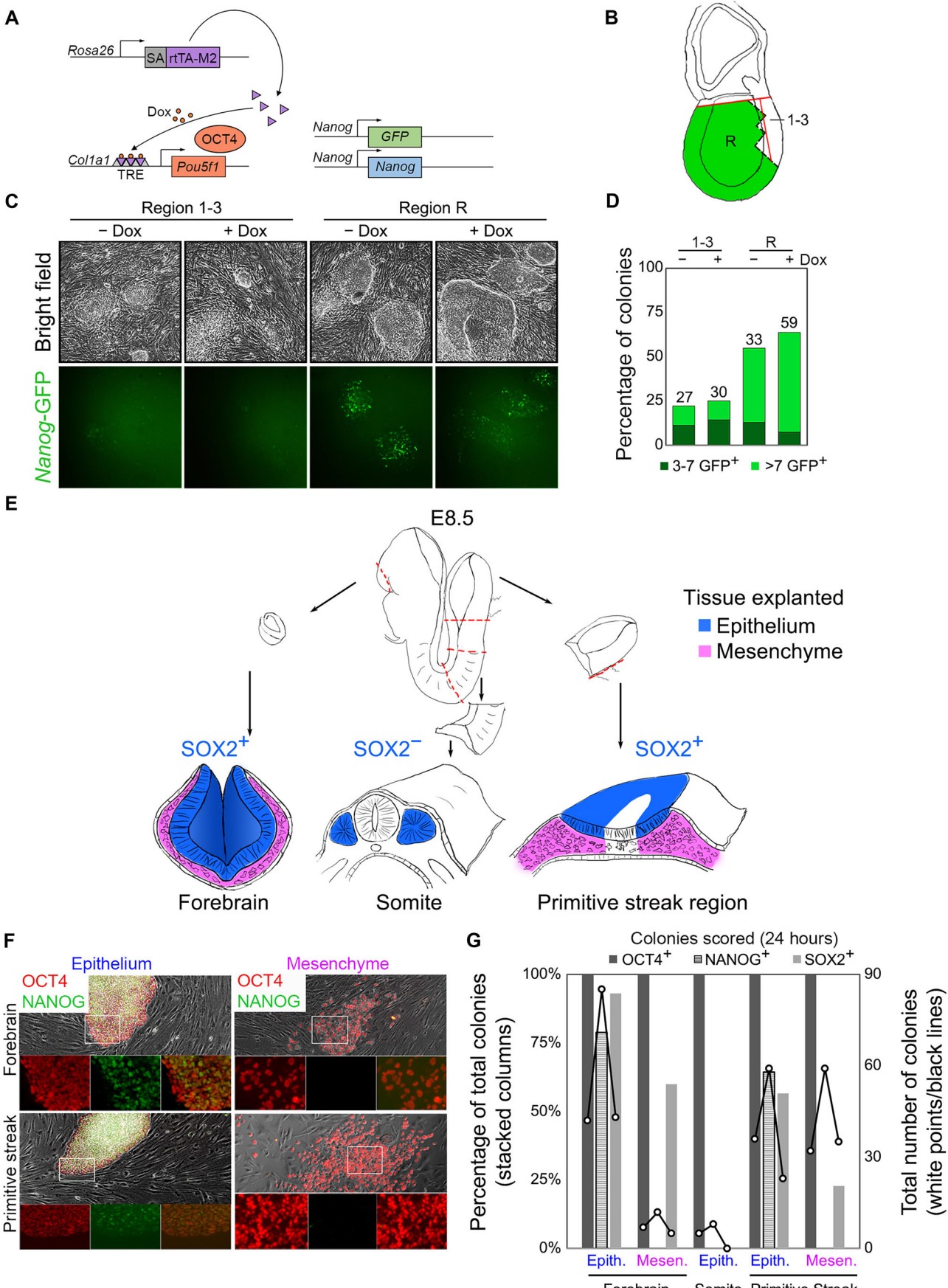

◀ **Figure EV5.  Enforced OCT4 expression does not restore pluripotency to posterior epiblast.**

(A) The doxycycline (Dox)-inducible OCT4 system. Ectopic *Pou5f1*/OCT4 expression is under the control of a tetracycline response element (TRE) in a transgene targeted to the *Col1a1* locus. The reverse tetracycline transactivator (rtTA) is constitutively expressed from the *Rosa26* locus after a splice acceptor (SA). One endogenous allele of *Nanog* coding sequence is replaced with GFP open reading frame (ORF). (B) Posterior (regions 1–3) and remainder (R) epiblast were explanted. Green domain is the pluripotent domain identified in Fig. 2F. (C) Representative GFP fluorescence images of explants at 24 h +/− Dox from regions 1–3 or region R. (D) Colony quantitation based on GFP expression level (Fig. EV4F). Total colony numbers analysed in 3 experimental replicates are indicated above each bar. (E) Forebrain, somites and primitive streak regions were dissected from 4 independent E8.5 somite-stage embryos (red dashed lines). The epithelium (blue) and mesenchyme (magenta) layers were separated in forebrain and primitive streak regions; in the somite region only the epithelial somites were explanted. (F) Immunofluorescence analysis of forebrain and primitive streak explants for OCT4 (red) and NANOG (green). Bright-field and immunofluorescence images are overlaid to show expression in explanted cells. (G) Colonies from each explant were scored for the presence of OCT4, NANOG or SOX2-expressing cells and are displayed as percentages of total colonies (stacked columns). The total colony number (sum of 2 experiments) is plotted on a secondary axis (black lines with white points). Source data are available online for this figure.

    

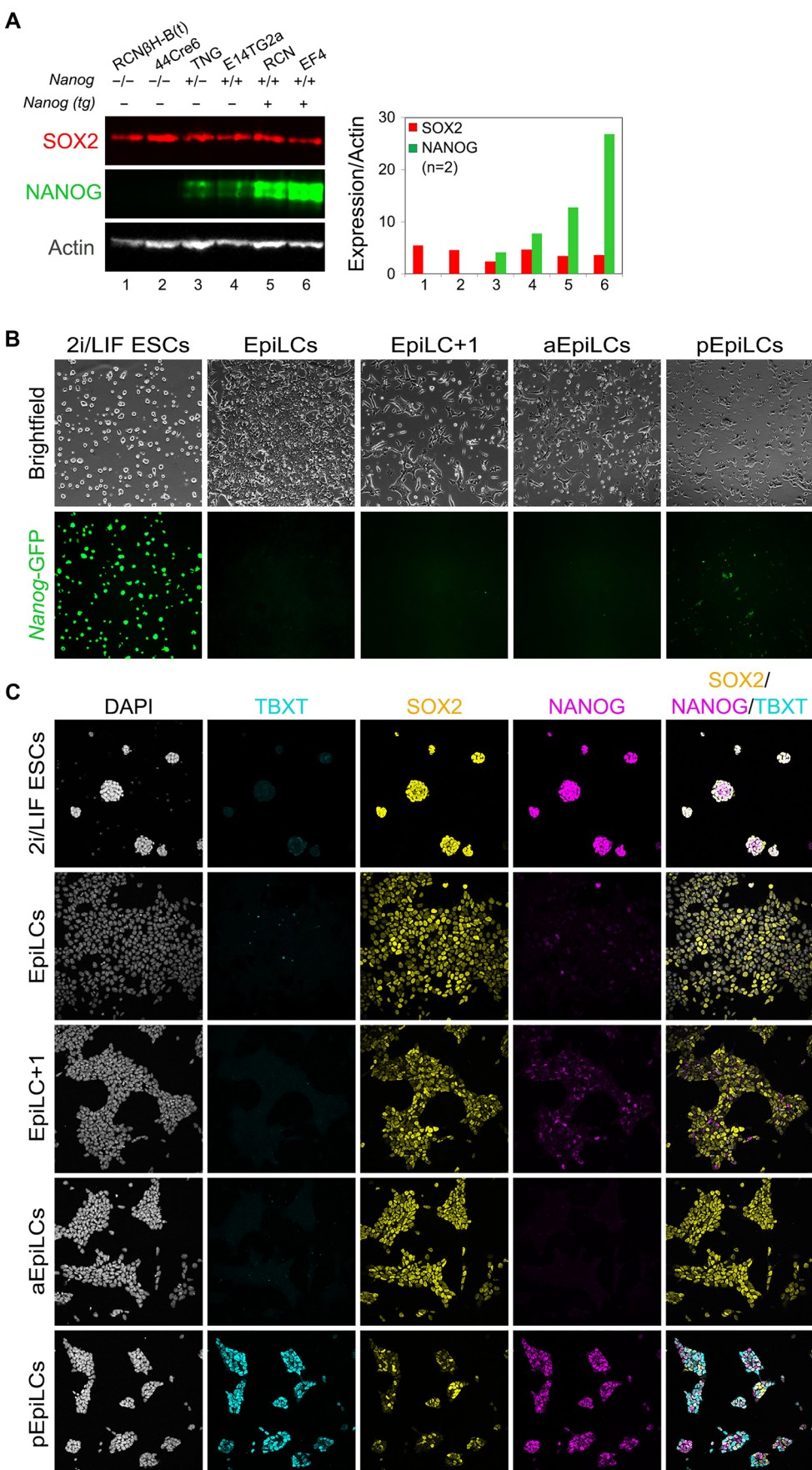

**Figure EV6.   Supporting Fig. 3.**

(A) Left, immunoblot analysis of SOX2 (red), NANOG (green) and β-ACTIN (grey) in ESCs overexpressing NANOG (EF4 and RCN), *Nanog*$^{+/+}$ (E14TG2a), *Nanog*$^{+/-}$ (TNG) and *Nanog*$^{-/-}$ (44Cre6 and RCNβH-(t)) ESCs. Right, quantification of SOX2 (red) or NANOG (green) band intensity relative to that of β-ACTIN. Shown is a representative of two independent replicate experiments. (B) Morphology and GFP expression of *Nanog*-GFP cells in the indicated states (see Fig. 3A). (C) Immunofluorescence of NANOG (magenta), SOX2 (yellow) and TBXT (cyan) in E14Tg2a cells in the indicated states (see Fig. 3A).

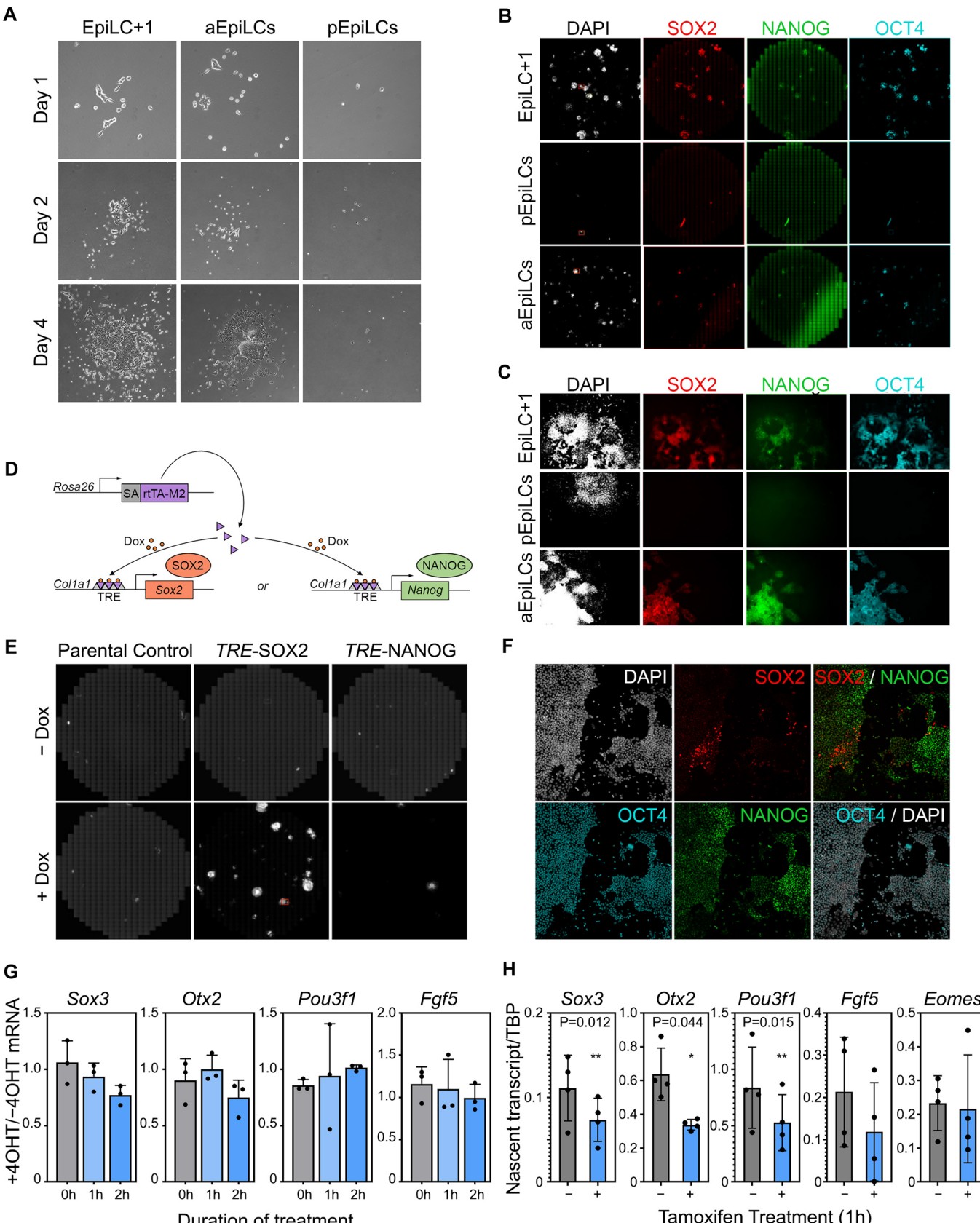

◀ **Figure EV7.  Supporting Fig. 3.**

(A) Typical colony morphologies during the EpiSCs colony-forming assay seeded from the three indicated starter populations (see Fig. 3A). (B) Immunofluorescence of SOX2, OCT4 and NANOG at day 7 of the EpiSC colony-forming assay showing a representative plate-scanned area of 9.62 cm². (C) High magnification view of the fields indicated by red squares in (B), showing SOX2 (red), OCT4 (cyan) and NANOG (green) fluorescence channels. (D) The doxycycline (Dox)-inducible SOX2 or NANOG system. Ectopic SOX2 or NANOG expression is achieved after recombination-mediated cassette exchange (RMCE) of a cargo vector was achieved in parental KH2 cells to deliver *Sox2* or *Nanog* sequences under the control of a tetracycline response element (TRE) in the *Col1a1* locus. The reverse tetracycline transactivator (rtTA) is constitutively expressed from the *Rosa26* locus after a splice acceptor (SA). (E) DAPI staining at day 8 of the EpiSCs colony-forming assay (see Fig. 3D) showing a representative area of 9.62 cm² for the indicated cell line without (−) or with (+) doxycycline. (F) High magnification view of the field indicated by a red square in (E), showing SOX2 (red), OCT4 (cyan) and NANOG (green) fluorescence channels. (G, H) Quantitative RT-PCR of *Sox3*, *Otx2*, *Pou3f1* and *Fgf5* total mRNA (G) or *Sox3*, *Otx2*, *Pou3f1*, *Fgf5* and *Eomes* nascent transcript (H) expression in *Nanog*$^{-/-}$; *CAG*$^{NanogERT2-IRES-puro}$ (NERT) EpiLCs treated with pEpiLC medium containing (+) or without (−) tamoxifen for the indicated hours. Expression levels are normalised to TBP (H) and the −4OHT control (G); values are means ± s.d.; points show values for three independent biological replicates.

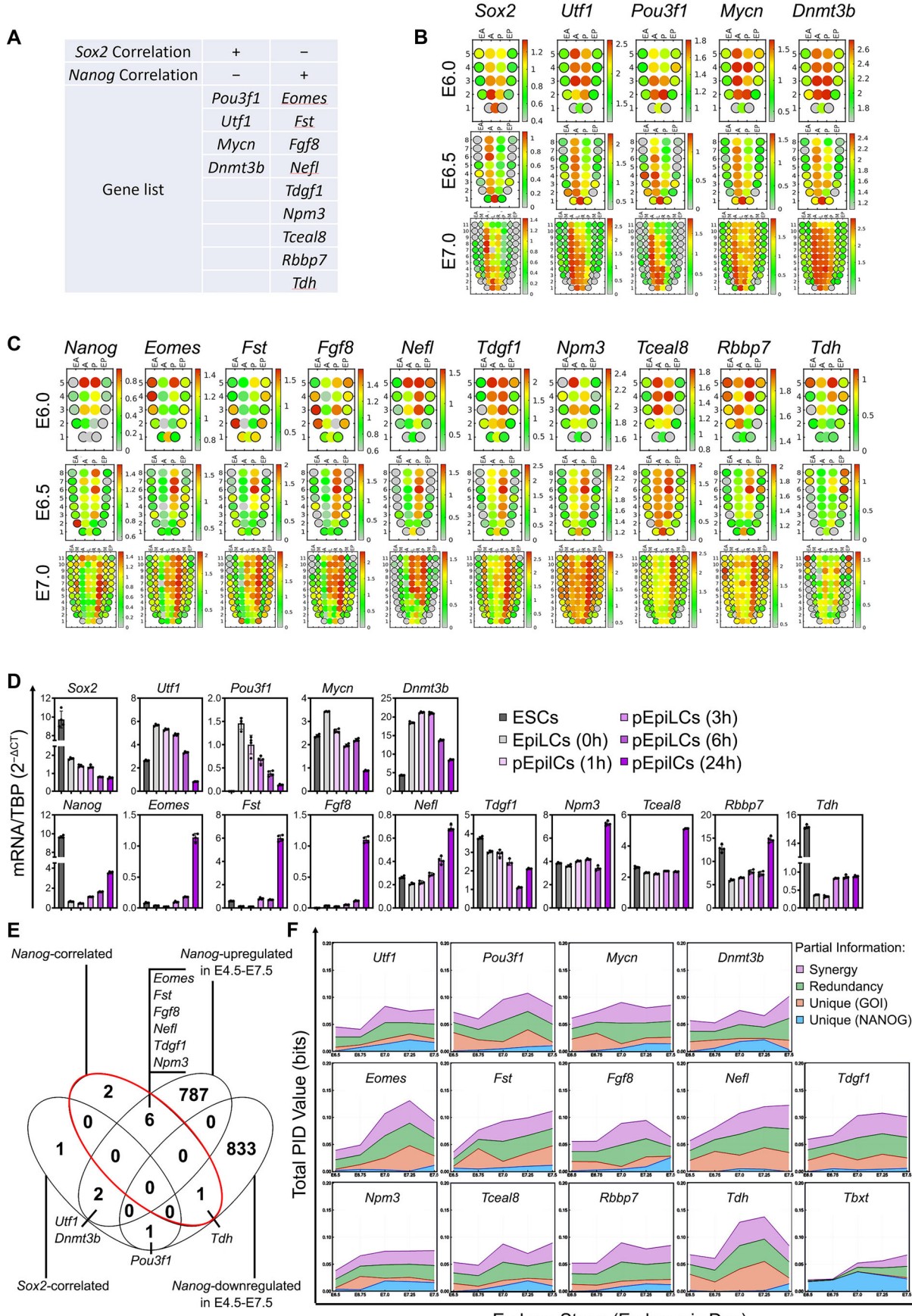

◀ **Figure EV8. Supporting Fig. 4.**

(A) Genes that show consistent opposite correlation between *Sox2* and *Nanog* across more than 1 stage of mouse embryos from E6.5 to E7.5; see also Dataset EV2 Genes that are positively correlated with *Sox2* and negatively with *Nanog* are denoted *Sox2*-correlated genes. Conversely, the opposite set of genes are *Nanog*-correlated genes. (B, C) Spatial transcriptomic 'corn' plots (of *Sox2* and *Sox2*-correlated genes (B) or *Nanog* and *Nanog*-correlated genes (C) from E6.0 to E7.0. Each dot represents the laser-captured microdissected section of the mouse embryo. Red is high expression and green is undetected expression. Corn plots were generated using the scGastrulation web portal as described in (Wang et al, 2023). (D) Quantitative RT-PCR of *Sox2*, *Nanog* and their respective-correlated genes in 2i/LIF ESCs, EpiLCs and EpiLCs cultured in pEpiLC condition for the indicated hours. Values are normalised to *Tbp* and presented as $2^{-\Delta CT}$. See Fig. 4B for expression normalised to EpiLCs. (E) Intersection between Nanog- or Sox2-correlated genes, and genes that change significantly upon Nanog overexpression between E4.5-E7.5 (Tiana et al, 2022) (F) Stacked plots of total partial information decomposition (PID) for predicting Sox2 expression levels in mouse embryos from E6.5 to E7.5. PID values that is contributed uniquely by NANOG (blue) or the indicated *Nanog*-correlated gene-of-interest (GOI, orange) is shown along with redundant (green) or synergistic (magenta) information conferred by NANOG and the indicated GOI for each embryo stage.

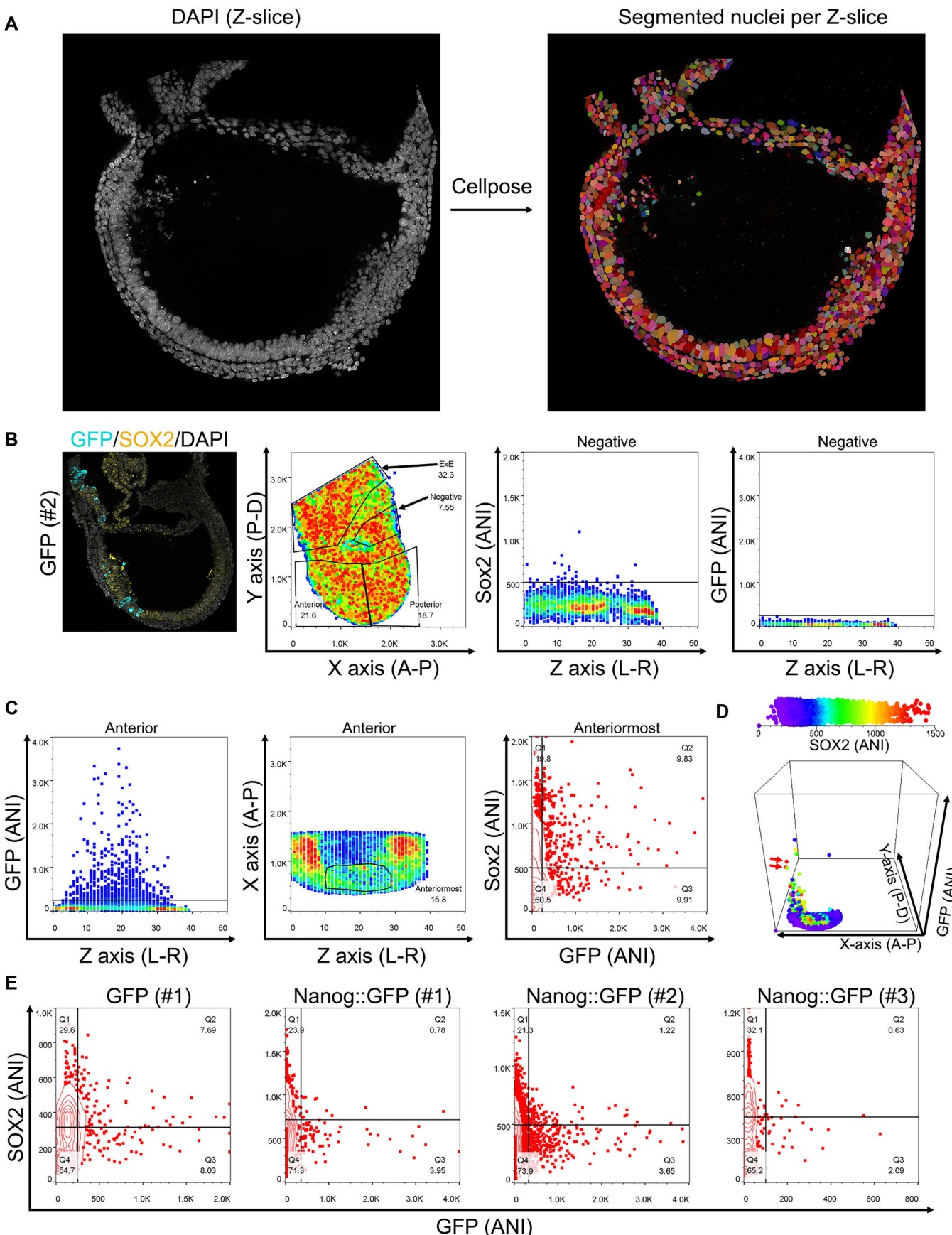

◀   **Figure EV9.   Supporting Fig. 5.**

(**A**) A representative DAPI-stained Z-slice in a confocal stack was segmented using a machine-learnt model Cellpose to generate segmented nuclear area per each Z-slice in the stack. (**B–D**) Example of workflow to extract Sox2 expression in Nanog-GFP or control GFP electroporated embryos with their neighbours using control GFP embryo 2. (**B**) A representative Z-slice near the midline of an electroporated embryo showing the expression of SOX2 (yellow), GFP (cyan), overlaid on DAPI (grey). The segmented nuclei of this embryo are plotted with its X/Y coordinates, representing the anteroposterior (A–P) and proximodistal (P-D) axes, respectively. Gates were drawn to extract the segmented nuclei in the anterior, posterior, ExE and an internal negative region. SOX2 and GFP ANI thresholds were determined using the negative region along the Z axis (corresponding to the left-right axis) to ensure the threshold captures any diffraction problems due to the thickness of the sample. (**C**) Segmented nuclei with high GFP ANI values were observed in the centre of the Z axis indicating anterior midline electroporation. The electroporated region is extracted to ensure that the SOX2 expression levels are comparable between GFP+ and GFP− cells. The relationship between SOX2 ANI and GFP ANI per segmented nucleus in this anteriormost region is represented in a 10% contour plot showing all events. (**D**) A 3D representation of the segmented nuclei in the embryonic (i.e. anterior + posterior) region plotted with GFP ANI in the vertical axis. The colour of each dot represents the detected Sox2 ANI value (top colour scale) to show that high Sox2 expression is permitted in high GFP-expressing cells without NANOG overexpression. (**E**) Similar contour plots showing the relationship between SOX2 and GFP ANI in another embryo electroporated with *CAG*-GFP in comparison to three independently electroporated embryos with *CAG*-Nanog::GFP.

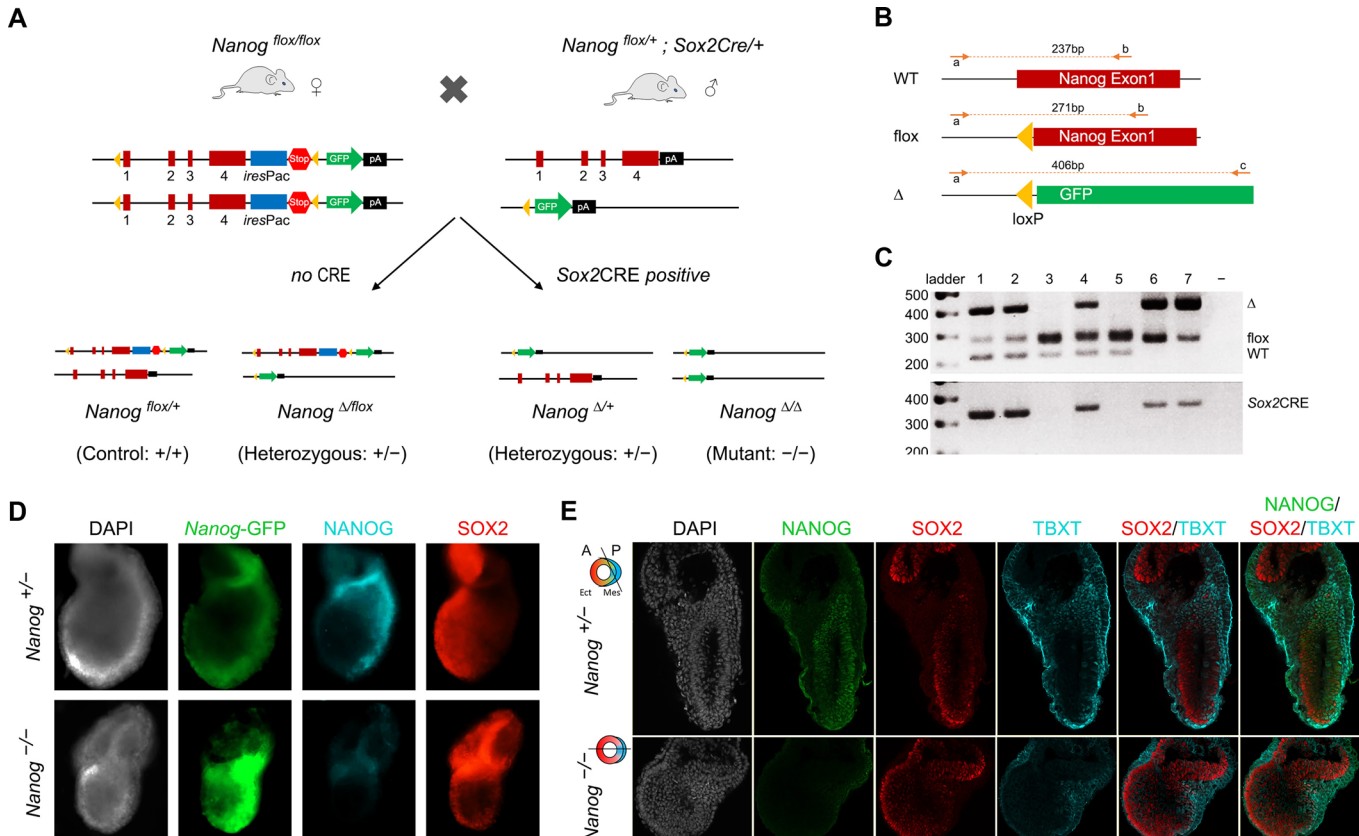

**Figure EV10.   Supporting Fig. 5.**

(**A**) Schematic of *Nanog*-conditional knockout by *Sox2*CRE. *Nanog*<sup>flox/flox</sup> female mice were crossed with *Nanog*<sup>flox/+</sup>; *Sox2*<sup>Cre/+</sup> male mice. As *Sox2*CRE is heterozygous, one in four offspring will be *Nanog* conditional knockout (*Nanog*<sup>Δ/Δ</sup>). (**B**) Genotyping strategy. Three primers, one forward primer (a) and two reverse primer (b, c) were used. "bp" denotes base pair. (**C**) Genotyping results of embryos derived from *Nanog*<sup>flox/flox</sup> females crossed with *Nanog*<sup>flox/+</sup>; *Sox2*<sup>Cre/+</sup> male mice. The extraembryonic regions were lysed for genotyping. Lanes 6 and 7, which have no "WT" band but have flox, Δ and *Sox2*CRE bands, correspond to homozygous *Nanog*-conditional null embryos (*Nanog*<sup>Δ/Δ</sup>). (**D**) *Nanog*-GFP (green) fluorescence, NANOG (cyan) and SOX2 (red) immunofluorescence in *Nanog*<sup>+/−</sup> and *Nanog*<sup>−/−</sup> embryos. The *Nanog*-GFP excised allele (details in **B**) is still responsive to signals restricting *Nanog*-GFP to the posterior region. (**E**) Representative longitudinal optical sections of *Nanog*<sup>+/−</sup> and *Nanog*<sup>−/−</sup> embryos. Schematic shows approximate plane of section on a notional transverse section of each embryo. NANOG (green), SOX2 (red), and TBXT (cyan) immunofluorescence showing TBXT is correctly localised to the primitive streak in SOX2+ cells.

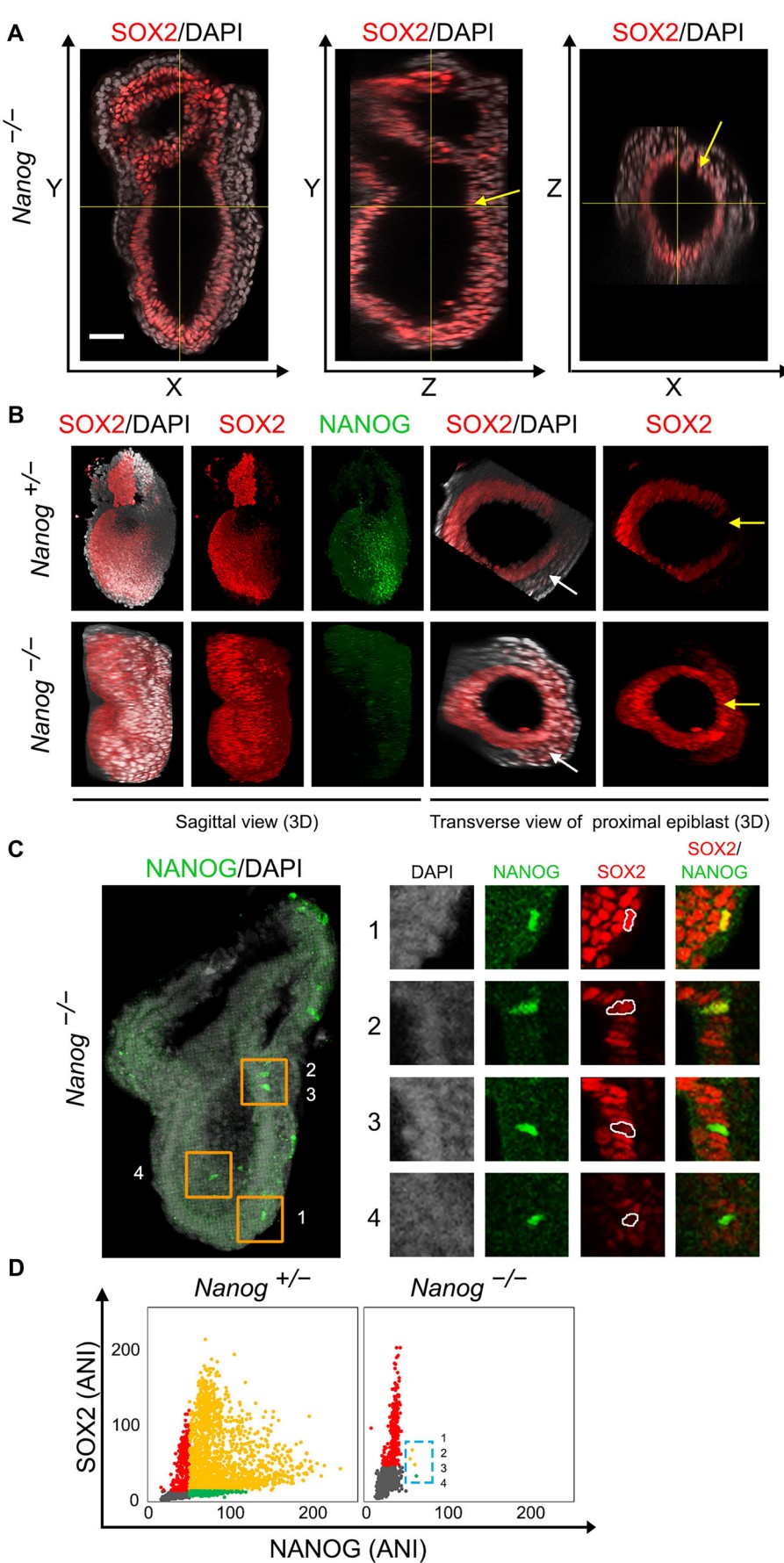

◀ **Figure EV11. Supporting Fig. 5.**

(A) Orthogonal view of a *Nanog*$^{-/-}$ embryo. Left, XY projection. Middle, YZ projection, approximating a sagittal section with anterior to the left. Righthand image, XZ projection, shows the posterior epiblast expressing SOX2 (yellow arrow). SOX2 = red; DAPI = grey; scale bar = 40 μm. (B) 3D rendering showing sagittal (left) and transverse (right) views of the *Nanog*$^{-/-}$ embryo in (A) along with a *Nanog*$^{+/-}$ embryo from the same dataset. The transverse view was generated from an equivalent cropped region of the proximal embryonic region to that shown in Fig. 5C. *Nanog*$^{+/-}$ embryos showed no SOX2 posteriorly, whereas SOX2 was detected in the posterior epiblast (yellow arrow) and in the ingressed mesoderm (white arrow) of *Nanog*$^{-/-}$ embryos. (C) Left, maximum Z projection of a segment of a second *Nanog*$^{-/-}$ embryo containing the location of the detected Nanog-positive cells (orange boxes, numbered). Right, each NANOG+ cell is shown overlaid on SOX2 expression. (D) Quantitation of SOX2 (red) and NANOG (green) ANI in a representative *Nanog*$^{+/-}$ or *Nanog*$^{-/-}$ embryo. A total of 3 *Nanog*$^{+/-}$ and 6 *Nanog*$^{-/-}$ embryos were analysed. Thresholds were gated using an internal negative region. The 4 detected Nanog-positive cells (blue box, numbered in order of SOX2 ANI level) had low or negative SOX2 ANI.

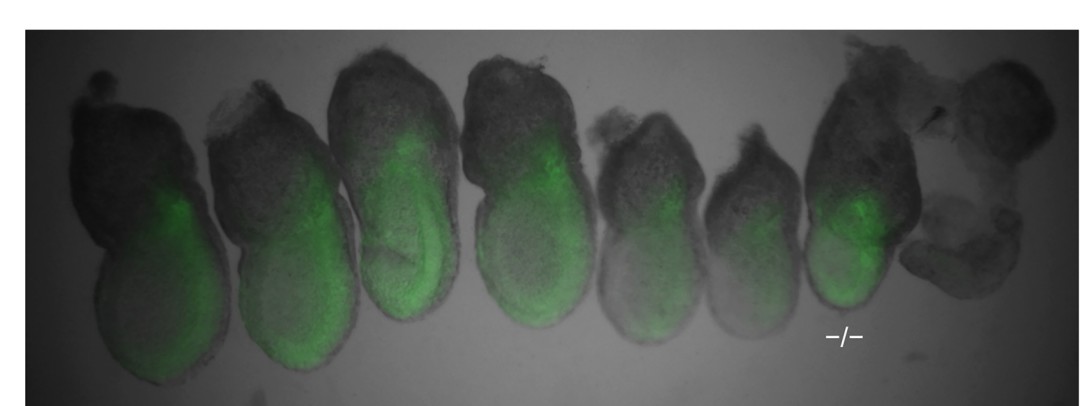

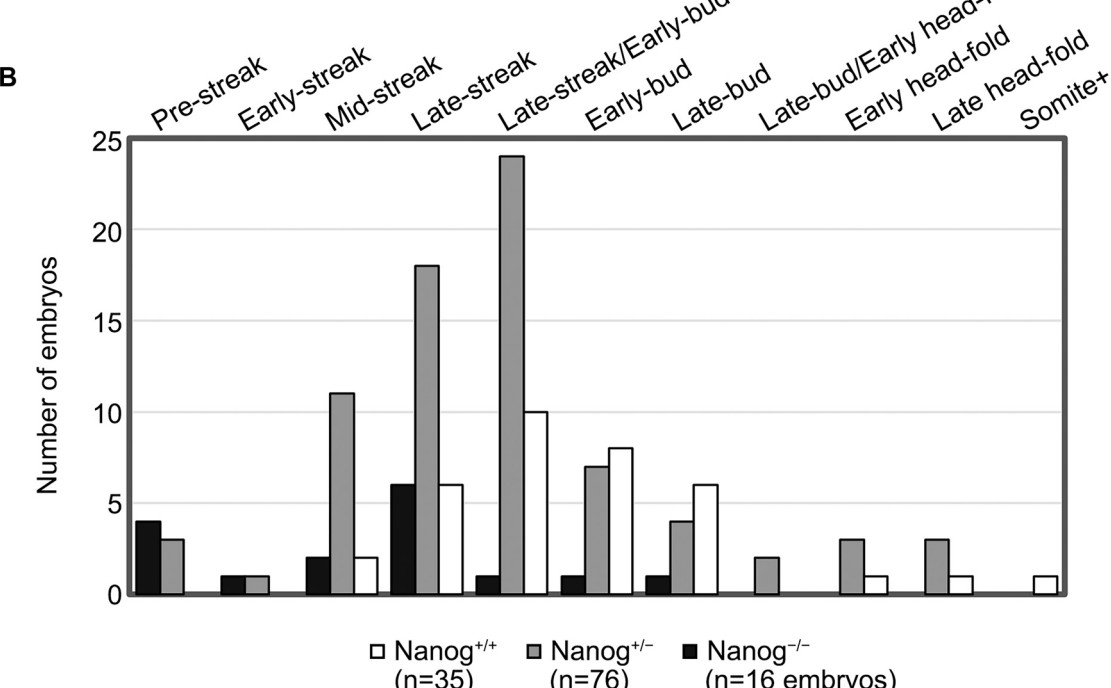

**Figure EV12.  Supporting Fig. 5.**

(**A**) Composite bright-field and *Nanog*-GFP fluorescence images of freshly dissected embryos at E7.5. *Nanog* genotyping (see Fig. EV11A for details) results are indicated below each embryo. (**B**) *Nanog*^+/+^ (white bars), *Nanog*^+/−^ (grey bars) and *Nanog*^−/−^ KO (black bars) embryos were scored by the morphological landmarks of the Downs and Davies staging prior further analysis. The total numbers of embryos analysed across all experimental replicates for each genotype are indicated.

