## [Peer Review File · The EMBO Journal]

NANOG is repurposed after implantation to repress Sox2 and begin pluripotency extinction

Frederick Wong, Man Zhang, Ella Thomson, Linus Schumacher, Anestis Tsakiridis, James Ashmore, Tong Li, Guillaume Blin, Eleni Karagianni, Nicholas Mullin, Ian Chambers, and Valerie Wilson

Corresponding authors: Valerie Wilson (v.wilson@ed.ac.uk) , Ian Chambers (i.chambers@ed.ac.uk)

Review Timeline:

Submission Date:	21st Mar 25
Editorial Decision:	11th Apr 25
Editor's Correspondence:	18th Apr 25
Editor's Correspondence:	24th Apr 25
Revision Received:	4th Jul 25
Accepted:	23rd Jul 25

Editor: Daniel Klimmeck

Transaction Report:

(Note: Please note that the manuscript was previously reviewed at another journal and the reports were taken into account in the decision making process at The EMBO Journal. With the exception of the correction of typographical or spelling errors that could be a source of ambiguity, letters and reports are not edited. Depending on transfer agreements, referee reports obtained elsewhere may or may not be included in this compilation. Referee reports are anonymous unless the Referee chooses to sign their reports.)

Referee #1 (Remarks to the Author):

In this manuscript “Nanog is repurposed to repress Sox2 and initiate developmental extinction of pluripotency,” the authors present results aimed at increasing our knowledge on the extinction of pluripotency. While some of the results are novel, I feel that the overall message is incremental in nature, with some of the results having been described before and while very interesting for the field, this manuscript would be more appropriate for a specialized journal.

We acknowledge that the original submission may have underrepresented the novelty of the result that NANOG suppresses postimplantation pluripotency. NANOG is pivotal to stabilising the preimplantation pluripotency gene regulatory network. Therefore our data showing the opposite is the case after implantation is, in our opinion, novel and of wide interest, pointing to a fundamentally different (and not simply destabilized) pluripotency gene regulatory network after implantation.

*The postimplantation segregation of SOX2 and NANOG has been reported previously. Importantly however, segregated NANOG and SOX2 expression has not previously been quantified in dual immunostaining through development, nor has the importance of this expression pattern been pursued. In our substantially revised version, we have now highlighted which aspects of the analysis in Figure 1 are novel and cite relevant previous publications (**lines 57-58**). We also include in the revision new data (**Fig 1c, d**) that shows definitively that NANOG and SOX2 negatively correlate before the onset of gastrulation, and thus the negative correlation cannot simply be a consequence of mesoderm lineage differentiation. The inclusion of new datasets (**new Fig. 3, 5 a,b and Supplementary Figures 6 b,c and 7, 9-10**), showing that NANOG inhibits Sox2 and thus dismantles pluripotency, strengthens our conclusions on this novel role of NANOG (see also response to reviewer 2 below, whose main criticism is that this claim is not supported by data). New transcriptomic data, combined with data mining and modelling (**new Fig 4a-c, f-h, and Supplementary Figure 8**), broadens this conclusion to indicate that Nanog is part of a broader gene regulatory network that prepares cells for early pluripotency loss and, thereafter, differentiation to mesoderm and endoderm.*

Specific comments

1. In Extended Data Figure 2a, the authors mentioned that Sox2 and Nanog are not well correlated in EpiSCs; rather they seem reciprocally regulated, at least in part because of Sox2 repression by Nanog. Is this phenomenon observed in human ESCs as well?

*Mouse EpiSCs and hESC have been reported to be in a similar state, different to mouse ESC (see Brons et al. DOI 10.1038/nature05950, Vallier et al. DOI 10.1242/dev.033951), Tesar et al. DOI 10.1038/nature05972, Greber et al. DOI 10.1016/j.stem.2010.01.003. However, to our knowledge SOX2/NANOG protein correlation/anticorrelation has not been directly compared under the same culture conditions. Whilst this maybe an interesting phenomenon, it is outside the scope of this study. Nonetheless, we have now included references in the discussion (**lines 359-360**) that provide evidence that NANOG and SOX2 inhibit the differentiation of neural and mesodermal lineages respectively in human ESC {Wang, Z., et al. [(2012).*

Distinct Lineage Specification Roles for NANOG, OCT4, and SOX2 in Human Embryonic Stem Cells. Cell Stem Cell 10, 440–454] and Vallier, L et al. (2009) DOI 10.1242/dev.033951}. Although these reports read out eventual differentiation outcomes, their data is consistent with the roles we identify for NANOG and SOX2 in mouse.

2. In Fig.3, forced Sox2 expression counteracts the loss of postimplantation pluripotent characteristics. More characterization of the transfected cells would strengthen the authors' claim, such as gene expression comparison in Region "R," Region "1" and Sox2-forced expressed Region "1") to see if Sox2 actually helps maintain a pluripotent state as Region R. Another experiment that is needed is teratoma formation testing by Sox2-forced expressing cells.

*Explanted cells from electroporated embryos can only transiently express SOX2, making this system unsuitable for addressing whether continuous SOX2 expression in the posterior epiblast can restore pluripotency. However to address this important question, we devised an assay in EpiLCs where we mimicked the formation of anterior (SOX-high, NANOG-low) or posterior (NANOG-high, SOX-low) epiblast by specific culture modifications and asked whether cells in these states could establish a pluripotent EpiSC phenotype. In anterior, SOX2-high epiblast-like cells, EpiSC colonies were readily produced. In posterior epiblast, EpiSC colonies could not ordinarily be produced. Crucially, we were able to rescue this pluripotency deficit in posterior epiblast-like cells in vitro by ectopic expression of SOX2. These new data are now presented in new **Fig. 3 and Supplementary Figs. 6b,c and 7.***

Induced Oct4 expression should be confirmed.

*The induction of Oct4 expression has been confirmed in many of our previous experiments. We draw the reviewer's attention to **Supplementary Fig. 5f** and our previous publications showing Oct4 induction in vitro and in vivo using this forced Oct4 expression mouse line (Osorno, Tsakiridis et al doi:10.1242/dev.078071; Economou et al DOI 10.1186/s12861-015-0084-7).*

3. In P8, why do the authors use Nanog -/- EpiLCs and not EpiSCs? EpiLCs and EpiSCs are different so the authors should not combine the results from EpiLCs and EpiSCs to generalize their claims.

*We apologise for not clarifying the rationale for using these cell types. EpiLCs and EpiSCs reflect aspects of the early and later postimplantation epiblast respectively. Differentiation of ESCs to EpiLCs is robust and relatively homogeneous. In contrast, while EpiSCs can be maintained indefinitely, they are more heterogeneous and subject to fluctuations in the relative proportion of subpopulations in culture (Tsakiridis et al, 2014, doi: 10.1242/dev.101014). Thus the EpiLC state, where NANOG is low and SOX2 moderately high, is an ideal starting point to examine how postimplantation Nanog induction affects SOX2. EpiSCs are used as an endpoint to demonstrate postimplantation pluripotent cell self-renewal. Crucially, we see anticorrelation in both cell types (**Fig 3, 4 and Supplementary Fig 3,6**) indicating negative regulation of Sox2 by Nanog (**Fig. 3g-i and Supplementary Fig. 7g,h**). We have now clarified the rationale for using EpiLCs (**lines 187-190**). While the conclusions from EpiSC data agree with those from EpiLC differentiation, we think that following*

EpiLC differentiation alone makes for a more linear narrative and therefore have removed data showing the downregulation of SOX2 after NANOG overexpression in EpiSCs. We are happy to re-include this data if the reviewer feels it is necessary.

4. Interestingly, heterozygous Nanog deletion induces Sox2 expression. How is the development of Nanog^{+/-} mice, altered, normal? And after they are born?

Nanog^{+/-} mice are viable and fertile. This is not unprecedented nor an argument against a role for NANOG in suppressing pluripotency. Indeed, mice heterozygous for a null Sox2 allele are viable and fertile, (Avilion et al). However, heterozygous Sox2 mutant cell lines are defective in neural differentiation. This defect can be rescued in vitro by BMP and Nodal signalling inhibition (Corsinotti, Wong et al. eLife; doi: 10.7554/eLife.27746), suggesting that the embryonic environment (e.g. factors secreted by visceral endoderm) may compensate for the dosage effects caused by heterozygous mutations.

Although Nanog^{+/-} cell lines do have higher Sox2 levels, consistent with negative regulation, we have now removed this data in favour of the more direct comparison of Nanog versus Sox2 levels following induction of Nanog by treatment of cells with, or without Nanog inducer (Fig. 3h,i).

5. On P8, the authors indicate that “Nanog negatively regulates Sox2 directly.” I totally disagree. First, I do not see that Sox2 expression is not affected by Tam treatment in Fig.4b. The negative effect on Sox2 is too weak. Is this result statistically significant? Second, there is no evidence for direct regulation. The authors should show Nanog localization on Sox2 promoter and conduct reporter experiments.

*This was a suggestion in our original manuscript. (p8 ‘... **suggesting** that Nanog negatively regulates Sox2 directly’). We have since found that both nascent (Fig.3i) and mature (Fig. 3h) Sox2 transcripts are downregulated in posterior epiblast-like cells (pEpiLCs; EpiLCs plated in Activin and CHIRON) within 1h of tamoxifen treatment of Nanog-ER^{T2} cells, increasing the likelihood that Nanog directly regulates Sox2. Both results are reproducible over at least 3 biological repeats and are statistically significant (p<0.05).*

Nevertheless, the reviewer’s comment that the negative effect on Sox2 is ‘too weak’ requires careful consideration. How weak is ‘too weak’? We have found that a threshold total SoxB1 complement is necessary to maintain EpiSC self-renewal (Corsinotti, Wong et al. eLife; doi: 10.7554/eLife.27746). By calculating the total SoxB1 mRNA levels, it appears that expressing as little as 17% of wildtype EpiSC SoxB1 levels permits some degree of self-renewal. This is consistent with the observation in this study that approximately 20% of maximum SOX2 immunofluorescence seems to be enough to initiate EpiSC pluripotency in explanted epiblast (Fig. 2f,g). Therefore an initial suppression of Sox2 to 50-70% of wildtype transcript levels is indeed weaker than predicted to ablate pluripotency. Therefore, although NANOG apparently represses Sox2 directly, this rapid repression may not by itself completely explain the disappearance of pluripotency.

Instead, in postimplantation cell types in vivo or in vitro, immunohistochemistry shows a more pronounced SOX2 protein reduction to near-background levels in the highest NANOG-expressing cells by 24 hours after the initial appearance of NANOG (Figure 1,4e, Supplementary fig 1,2). That this reduction of SOX2 is caused by Nanog is shown in Fig. 5b, where SOX2 is reduced in NANOG-expressing cells to under ~25% of the levels in neighbouring cells. Therefore Nanog is unlikely to act by simply suppressing Sox2 transcripts. Figure 1d shows that lateral regions of the epiblast express intermediate levels of both proteins. This correlates very well with the map of pluripotent regions of the epiblast generated by microdissection and either explant culture (Figure 2f) or teratoma formation (Figure 2c), such that low-Nanog, intermediate Sox2-expressing cells in the lateral epiblast are pluripotent, while only the posterior extreme of the epiblast, expressing high Nanog and undetectable Sox2 is not pluripotent.

Our conclusions would not be altered whether this regulation was direct or not. We believe that the detailed characterisation of transcriptional and post-transcriptional regulation would be more suitable for a separate paper. We have acknowledged and discussed the modest but significant reduction in Sox2 mRNA levels prior to wholesale disappearance of Sox2 protein (lines 217-227, 337-339).

6. On P10, L, "Therefore, the PGRN activity of naïve and primed states appears to be fundamentally distinct." Is it known? If so, please rephrase.

We apologise for the lack of clarity. In theory, the reduction in Sox2 and Nanog levels in EpiSCs compared to ESCs could simply result from a destabilised PGRN. In this scenario, 'naïve' pluripotency markers such as Esrrb, Rex1 and Klf4 could be absent because factors at the periphery of the PGRN are not maintained in EpiSCs, and early lineage differentiation markers (e.g. T, Foxa2) might simply signify the presence of contaminating differentiated cells. The present work shows that the differences between naïve and primed states are instead rooted in a fundamentally distinct PGRN in which a core component of the naïve state, Nanog, is detrimental to the primed state. In support of this idea, others previously have shown distinctions between pre- and post-implantation pluripotent phenotypes that could not result from a merely weakened network. These include a switch from LIF to Activin/Fgf factor dependence (Tesar, 2007) and differences in Oct4 enhancer use (Tesar 2007, Yeom et al. 1996). Therefore, taking these points into account, we have completely rewritten the relevant paragraph (lines 318-329).

7. It would be interesting to try to force Nanog expression and determine whether it will facilitate extinction of postimplantation pluripotency.

See response to point 5 above. We show that forced Nanog expression in posterior epiblast-like cells cannot rescue pluripotency (Fig.3f). We include new data that shows forced NANOG expression inhibits Sox2 in the anterior epiblast ≤6 hours after electroporation of Nanog (Fig. 5a,b; Supplementary Fig.9)

8. On P5, the authors mention that decreasing Sox2 expression at 24-48h in ESCs contributed to exit of pluripotency. However, in Extended Data Fig 3a-d, Sox2 expression does not decrease that much. Are there significant differences between at 24 and 48h?

This refers to data measuring the loss of naïve pluripotency by replating differentiating ESCs in LIF, where we stated “A decrease in Sox2 at 24-48h marked the transition point when most cells lost the capacity to form self-renewing ESC colonies when replated in LIF/FCS (Supplementary Fig 3e), suggesting that decreasing Sox2 protein levels may contribute to exit from naïve pluripotency in vitro”. This has been removed from the manuscript as the effect of SOX2 levels on exit from naïve pluripotency have now been reported more fully in Corsinotti, Wong et al. eLife; doi: 10.7554/eLife.27746.

In Fig.3e.f, the authors claimed that restoring Nanog expression by forced Sox2 was sufficient to restore pluripotency, which is not consistent with what is shown in Extended Data Fig 3. Please explain.

*Fig 3e,f (new Fig 2i,j) examines the effects of enforcing SOX2 expression on proximal posterior epiblasts taken for explant in Activin/Fgf. **Supplementary Fig 3** describes changes in expression of Sox2, Nanog, Oct4 during ESC differentiation. These are fundamentally different. In ES cells undergoing in vitro differentiation to EpiSCs, Sox2 mRNA and protein levels reproducibly decrease to around 20% of their ESC level (n= >3). Please see the answer to point 5 for a response to the question of the level of Sox2 required for postimplantation pluripotency.*

We agree that it seems counterintuitive to use Nanog-GFP expression as a marker of pluripotency when we argue that Nanog suppresses pluripotency. We acknowledge that the brief description in the original manuscript led to a lack of clarity. This paradox can be resolved because (1) A spike of Nanog-GFP (and NANOG) is activated by Activin/Nodal signalling, and predicts the competence of cells to form primed pluripotent cell lines (Osorno et al. Development 2012) (2) intermediate (but not high) Nanog levels, which are seen in EpiSC lines, are compatible with Sox2 expression and pluripotency.

*We now cite Osorno et al (2012) to explain the rationale for assaying Nanog-GFP in response to Sox2 electroporation in vivo (**lines 125-127**)*

9. The authors show that Sox2/Nanog expression was shifted during early embryogenesis and claim that this positive correlation was lost at E6.5. I have some concerns that I would like the authors to clarify. From the inducible Nanog experiment and the deletion study in Fig.4, t Nanog might negatively regulate Sox2 expression mildly at that stage. However, I am confused with what is shown in figure Fig4a. The authors suggest that Nanog expression wasn't affected by Sox2 deletion, but Sox2 expression was increased by Nanog deletion. However, the ratio looks similar in both experiments. The authors should repeat these experiments and statistically prove their claims.

*There is a very small increase in Nanog levels in cells lacking one copy of Sox2. In the revised version of this manuscript, we have decided to remove EpiSC Western blot data, together with immunofluorescence on independently-generated EpiSC lines where the levels of Nanog or Sox2 are altered by knockout or gain of function. Although this data is reproducible and supports negative regulation of Sox2 by Nanog, we think that the new data on EpiLC differentiation where Nanog is induced (**Fig. 3f**) is stronger, as we compare the levels of Sox2 and Nanog within a single cell line, eliminating any unpredicted cell line variation. We consider that*

the removal of the above data creates a more linear narrative. However we can reinstate the EpiSC data if the reviewer feels it would give a more complete picture.

10. In Fig. 4, the authors mention that Sox2 expression is reduced by inducible nuclear Nanog activity in Nanog^{-/-}EpiLCs, but on the other hand, Sox2 is repressed by Nanog overexpression in EpiSCs. Feedback mechanisms, as the authors mention, is not sufficient to interpret this result.

There seems to be no issue to address here, as the two statements agree.

11. The 3D data in Fig 4c and Extended Data Fig 9 is unclear and difficult to see.

The data in the righthand panels of Fig 4c (new Fig 5c) looks more blurred than the Z-section in the second column (labelled Sox2/DAPI) as it is a virtual section of a 3D reconstruction. However, the reconstruction and the virtual sections themselves are unambiguous. They show that in Nanog deleted embryos expression of SOX2 is abnormally extended, not only into the most posterior epiblast cells as they enter the primitive streak but also to cells in the emergent mesoderm. This indicates that removal of Nanog causes Sox2 expression to be retained in the posterior epiblast in vivo. Posterior Sox2 expression is observed in 6/6 Nanog homozygous null embryos. In contrast, posterior Sox2 expression is never seen in heterozygous or wildtype E7.5 embryos (0/11) (Figure 1, Figure 5c, Supplementary Figure 1, 10d-e, 11b). Four of the 6 imaged Nanog null embryos are presented (Figure 5c, Supplementary Figures 10d-e, 11a-d).

The experimental design of Nanog deletion by Sox2Cre might not be the most appropriate one since Nanog deletion might affect the development of early embryogenesis as seen in Fig. 4c.

We think that Sox2Cre is an appropriate tool to excise Nanog early post-implantation. Sox2Cre activity starts from early post-implantation stages, after pre-implantation Nanog expression has disappeared but before it is re-expressed post-implantation (Hayashi 2002 DOI: 10.1016/S0925-4773(03)00099-6). This experiment is designed to prohibit the subsequent embryonic re-expression of Nanog at the posterior epiblast prior to the onset of gastrulation.

I suggest Sox2CreERT as an option that may serve well for the authors to prove their message.

Although an inducible Cre transgene would be ideal, tamoxifen treatment during the peri-implantation period necessary to achieve this could induce diapause (Batlle-Morera et al. 2008 doi.org/10.1002/dvg.20442). Also, our experience and that of Danielian, et al (1998) Curr Biol 8, 1323–1326 is that excision of floxed cassettes by tamoxifen induction of Cre activity occurs with low efficiency in early postimplantation embryos.

12. In Fig1 and Ex. Fig 2, 3, the authors concluded that Nanog and Sox2 protein segregate to different regions of post-implantation embryos and different cell-populations derived from early embryo based on the result of fluorescence intensity quantification. I am concerned whether this sole image analysis might be enough. To further confirm their results, the authors should also use other approaches. Cell sorting and single cell analysis will strengthen the authors' conclusions.

To verify the differential localisation of Sox2 and Nanog in our specific regions of interest, we had carried out qPCR of region 1 and R (Figure 2b, Supplementary Fig 4c). As discussed above there are several published analyses showing distinct localisation of Sox2 and Nanog mRNA and protein to anterior and posterior respectively, which we have now referenced more fully. We have now mined the scRNA-seq data from Pijuan-Sala et al. (2019) and Peng et al. (2019) and include this analysis in Figure 4f-h and Supplementary Fig. 8 to show distinct sets of genes correlate with either Sox2 or Nanog in the epiblast and show that these genes are co-regulated in posterior epiblast-like cells in vitro. We also show that Nanog-correlated genes overlap with those from Tiana et al. (2022) that are upregulated following Nanog induction in vivo (Supplementary Fig 8e), and that these genes predict SOX2 localisation along with NANOG better, and earlier, than TBXT (Supplementary Fig 8f) We have also performed RNA-seq on posterior and anterior epiblast-like cells, which demonstrates that this system mirrors in vivo epiblast differentiation, where Sox2 and Nanog are induced in posterior and anterior conditions respectively. This data is presented in Figure 4a-c.

13. Other groups (see for instance Osorno et al, Development 139 (2012) 2288-98) have indicated that Nanog is not required for post-implantation pluripotency. The authors' conclusion rely on Nanog:GFP expression as an indicator of pluripotency. As indicated above, a comprehensive mRNA expression analysis will help to prove authors' claim.

Our previous demonstration that Nanog is not required for postimplantation pluripotency in EpiSCs (Osorno et al 2012 DOI 10.1242/dev.078071) does not preclude using the Nanog-GFP reporter as an indicator of EpiSC pluripotency. In response to point 8 above we discuss the apparent discrepancy between our statement that Nanog suppresses pluripotency and our use of the Nanog-GFP reporter as a proxy for EpiSC line derivation (also used in Osorno et al., 2012 where we showed that Nanog-GFP expression is Activin-dependent).

In the above study we also showed that Nanog null EpiSC lines can be derived from embryos and ESCs. This argues against a requirement for Nanog more convincingly than transcriptome analysis of single explanted cells that reactivate Nanog-GFP. In the present study, we argue that Nanog expression decreases pluripotency through negative regulation of Sox2.

14. Throughout this manuscript, statistical analyses are missing.

Statistical analyses have been included where relevant. The figure legends now include the number of biological repeats.

Referee #2 (Remarks to the Author):

Uncommitted epiblast cells in post-implantation mouse embryos progressively acquire the expression of lineage-specific markers before committing to specific lineages. The process of lineage specification is controlled by locally activated diverse signaling pathways and transcriptional regulators, but it remains obscure when and how individual cells become committed to specific lineages and lose multi-lineage differentiation capacity. The authors previously showed that a pluripotent cell population, evaluated by its ability to give rise to colonies of pluripotent epiblast stem cell (EpiSCs) in vitro, persists until the somite stage (Osorno et al., 2012). In this manuscript, Wong et al. extend the findings of the previous study by focusing on two transcription factors, Sox2 and Nanog, and provide insight into when and how pluripotency is first lost in mouse embryos. The authors found that proximal posterior epiblast cells, which first lose the expression of Sox2, are unable to give rise to EpiSC colonies. Based on reciprocal expression of Sox2 and Nanog, and the functional assays including overexpression of Sox2 and knockout of Nanog, the authors conclude that repression of Sox2 by posteriorly localized Nanog causes loss of pluripotency. Although the manuscript is well presented, the main claim that Nanog represses Sox2 and leads to lineage restriction is not supported by strong experimental evidence. In addition, from the aspect of developmental biology, the assessment of developmental potency needs a more careful investigation.

We focus here on the role of Nanog in suppressing postimplantation pluripotency, rather than addressing lineage restriction per se. We consider the inappropriate and persistent expression of SOX2 upon Nanog deletion in vivo (Figure 5c) is strong evidence for this claim. We have now included data showing that induced expression of Sox2 confers EpiSC identity on Nanog-high posterior epiblast-like cells (Figure 3d-f), and ectopic anterior expression of Nanog reduces Sox2 in vivo (Figure 5a-b). The reviewer has some issues with the data in the original submission which we answer below.

(1) Reciprocal expression of Sox2 and Nanog in post-implantation epiblasts is not a novel finding, but rather a well-described one in previously published papers (e.g., Hoffman et al., 2013; Sumi et al., 2013).

We had acknowledged previously published data on this point and did cite the Hoffman 2013 study (line 57-58) but have now also reworded this for clarity and now include a discussion of Sumi et al. 2013 (lines 342-348; see also our response to the point below). See also our response to point 8 from reviewer 1. While the reciprocal expression of Sox2 and Nanog in vivo has been described before, and reciprocal expression of the mRNAs have been described in vitro (Trott and Martinez Arias, 2013 DOI 10.1242/bio.20135934), SOX2 and NANOG proteins have not been quantified at single cell level over time, and the negative correlation has not been pursued further.

In the current manuscript, it is unclear how important the antagonism between Sox2 and Nanog is for lineage restriction. It is well established that localized activation of signaling pathways is responsible for the regionalization of the epiblast. There are several reports showing that Nanog is mainly induced by Activin/Nodal and WNT signals, which are active in the posterior proximal epiblast. The data shown in

Figures 4a and 4b do not provide convincing evidence to support the notion that Nanog represses Sox2. In Figure 4a, Nanog^{+/+} and Nanog^{+/-} are not compared on the same blot, and the difference at the transcriptional level is not shown.

Our manuscript primarily investigates loss of pluripotency rather than lineage restriction. Although the concepts are related, our assays are directed towards measuring pluripotent phenotypes, rather than differentiation potential. Indeed WNT stimulation leads to low SOX2/high NANOG expression (Sumi et al. 2013) while its inhibition has the converse effect. However expression of NANOG in the low-WNT anterior epiblast results in SOX2 downregulation (Fig. 5, this manuscript), suggesting that NANOG and WNT act in parallel to downregulate SOX2. Trott and Martinez Arias (2013) show that in ESCs differentiated in N2B27, Sox2 and Sox1 mRNAs correlate well and segregate from cells expressing Nanog and Tbra, providing an interesting parallel with our results in EpiLCs and embryos. Moreover, these authors show that Nanog is required to allow CHIR-mediated Wnt stimulation to efficiently block neural Sox1 expression and induce T-positive primitive streak phenotype. We now cite this study in our discussion on Wnt signalling and thank the reviewer for the prompt to include a discussion of this interesting point (lines 331-357).

The last point is incorrect. They were compared on the same blot, which was shown in its entirety in the supplementary data. However, we have now removed this data as the evidence provided by induced Nanog expression in postimplantation pluripotency strongly supports a negative relationship at the protein level, and does not carry the caveat that the different cell lines may have accumulated culture adaptations (See response to point 9 from reviewer 1).

In Figure 4b, the reduction of Sox2 transcript level is minuscule.

Our new data indicates that reduction in Sox2 mRNA levels to 70% within 1 hour of Activin+Chiron treatment is reproducible over 3 experiments, and a 50% reduction in nascent transcripts is reproducible over 4 experiments (Fig. 3h, i). Our previous data described a similar reduction in EpiLCs within 2 hours in Activin+Fgf. The reviewer's point that the reduction may not be enough to explain loss of pluripotency is discussed in our response to a similar point (#5) made by reviewer 1. Briefly, the reduction at the protein level after 24 hours to near-background levels in the highest Nanog-expressing cells in vitro (Figure 4e), and those at the posterior of the embryo (Figure 1b), is a strong enough effect to explain the loss of pluripotency.

In Figure 4c, Sox2^{cre}/Nanog^{f/f} embryos look developmentally delayed. Are markers for primitive streak and mesoderm expressed properly?

Nanog-deleted embryos tend to be delayed relative to heterozygous and wildtype littermates, although the range of stages overlaps with heterozygotes and wildtypes. We clarify this in new Supplementary figure 12b. Nanog^{-/-} embryos form mesoderm which expresses T, a primitive streak/nascent mesoderm marker (see Figure 5c and Supplementary figures 10e and 11a-b) and show correct localisation of the Nanog-GFP transgene (Supplementary figure 10d,12a). The negative regulation of Sox2 by Nanog predates the

formation of mesoderm (Fig 1a-d). Therefore the posterior expression of Sox2 in Nanog conditional null embryos is not simply because embryos are retarded and fail to generate a primitive streak.

The biological significance of Nanog-mediated Sox2 repression is also not clear, as Nanog-knockout embryonic stem cells (ESCs) and EpiSCs contribute to all somatic lineages according to the previous studies. The authors' model is also not consistent with the observation by Hoffman et al. that constitutive expression of Nanog by a transgene throughout post-implantation development does not affect the anterior localization of Sox2.

It is true that Hoffman et al. did not find changes in Sox2 localisation after Doxycycline-induced Nanog overexpression in embryos. However the number of nuclei (~100) and embryos (2 x Nanog-overexpressing, early-streak stage) analysed was relatively small, suggesting that a negative relationship could have been missed. Additionally, these authors did not report dual Sox2/Nanog immunostaining results, and did not comment on an apparently lower number of Sox2 positive nuclei in Nanog-overexpressing versus wildtype embryos (Hoffman et al Figure 2L,P versus Figure 1 D,I).

(2) The embryo explant experiments suggest that the posterior proximal region first loses the ability to give rise to EpiSC colonies. One concern is about the identity of the explants. Are they epiblasts? Were mesodermal and endodermal layers removed? What is the cell of origin of the EpiSCs? The same also applies to the explants of somite-stage embryos (Figure S5). What is the origin of the EpiSC colonies? Are Oct4 and Sox2 expressed in the original cells?

*We draw the reviewer's attention to **Supplementary fig 4e**. Dil labelling of the whole epiblast demonstrated that only explanted epiblast cells gave rise to (Nanog-GFP+) EpiSC colonies. Although epiblast was present in the proximal posterior region, it did not maintain Nanog-GFP expression in culture. We have now included this data in a main figure as it is important to the later arguments that Nanog is not simply forcing mesoderm differentiation.*

*As regards somite-stage embryos, we and others have shown that endogenous Oct4 is not expressed but that induced Oct4 expression only resulted in Nanog:GFP activation in explants from cells that express Sox2 (**Supplementary figure 5e-g**) (Osorno, Tsakiridis et al doi:10.1242/dev.078071; Economou et al DOI 10.1186/s12861-015-0084-7).*

Another concern is about the evaluation of pluripotency. The EpiSC-forming ability is just one criterion for pluripotency, and EpiSCs are required to possess the ability to self-renew under non-physiological culture conditions. It is likely that the posterior proximal region contains a high proportion of committed mesodermal cells and uncommitted yet mesoderm-primed brachyury-positive epiblast cells that are not expandable under EpiSC conditions.

We are unsure what the reviewer's concern is here. We used both EpiSC derivation and teratoma-formation as independent assays of pluripotency, which both identified the posterior proximal region as lacking pluripotency. This region contains cells fated for mesoderm, and together with our demonstration that they

are not pluripotent, it is reasonable to believe that posterior proximal epiblast is committed to mesoderm formation. Interestingly, TBXT expression itself is compatible with EpiSC maintenance (Tsakiridis et al, 2014, doi: 10.1242/dev.101014), although we did show in that study that TBXT+ EpiSC are biased towards mesoderm formation. The above considerations agree with our demonstration that the inhibition of Sox2 by Nanog is part of the mechanism by which cells restrict potency in order to commit to mesoderm fates.

It is also expected that loss of Sox2, but not Nanog, impairs the derivation of EpiSCs, as Oct4 and Sox2, but not Nanog, are absolutely required for the self-renewal of pluripotent stem cells in vitro.

The dispensability of Nanog for pluripotency (either in ESCs or EpiSCs) is not especially surprising. What is unexpected is that Nanog actively suppresses pluripotency. On a similar note, our data showed that unexpectedly, Sox2 (but not SoxB1) activity is dispensable for EpiSC self-renewal (Corsinotti, Wong et al. eLife; doi: 10.7554/eLife.27746).

Teratoma formation also requires cellular proliferation. Indeed, the posterior proximal region did not generate even mesodermal tissues (Figure 2e). Is it possible that uncommitted epiblast cells from different regions are exposed to growth factors that direct the differentiation into specific cell types? Or were these cells transplanted into different regions of embryos? It was shown that the posterior proximal epiblast can generate surface ectoderm if it is transplanted into the anterior epiblast region (Beddington, 1982).

This is an interesting question. Here, the reviewer suggests that region 1 epiblast is sensitive to the kidney environment, into which epiblast cells are transplanted, and does not proliferate/differentiate, whereas the rest of the epiblast responds by proliferating, self-renewing and differentiating. The fact that the proximal posterior region can proliferate and differentiate in vivo and when transplanted elsewhere in the epiblast indeed suggests that the kidney environment is unfavourable for growth of this region, whereas it is favourable for pluripotent cells. However we do not see that performing transplants into other embryos or adults would substantially alter the main conclusions of our manuscript.

Minor points:

In Figure 2f: Not only percentages, but also exact numbers of colonies should be shown.

The number of colonies is shown above the bars in Figure 2f. We can adapt the presentation of Figure 2f or the legend if the reviewer can specify why it was unclear in the original version.

In Figure 3f: The expression of the Sox2 transgene seems to be at super-physiological levels compared with the expression of endogenous Sox2 in colonies from region 8. Are these cells expandable like EpiSCs?

See response to reviewer 1 point 2. Extended culture was not possible with these cells; however our new data (Figure 3f) shows that induced Sox2 expression allows EpiSC colony formation in replated posterior epiblast-like cells.

In Figure 4: It is unclear which cell lines are used. Are these ESC-derived EpiSC-like cells? Do the heterozygous and homozygous lines in Fig. 4a originate from the same parental line? Were the Sox2 and Nanog lines in Fig. 4a cultured side by side?

The cell lines used were described in the materials and methods section. All EpiSC lines were derived in vitro from ESCs and cultured in identical conditions. However, we acknowledge that these independently-generated genetic modifications have different origins and potential culture adaptations. As stated in response to Reviewer 1 point 9, we have decided to remove this data in favour of the more straightforward demonstration that induced Nanog expression in EpiLCs leads to reduced levels of Sox2.

In Figure 4c: Do Sox2^{cre}/Nanog^{f/f} embryos develop further without any abnormality?

*There is some loss of Nanog homozygotes prior to E7.5, evidenced by a lower proportion of recovered mutant embryos. Intriguingly, we have occasionally recovered adults containing cells of Sox2^{cre}/Nanog^{f/f} genotype from these crosses. As the reviewer notes earlier, Nanog null ESCs can contribute to most cell types in chimeras (Chambers et al. Nature 2007 DOI: 10.1038/nature06403), suggesting that null cells are capable of normal development in presence of wildtype cells. As shown in **Supplementary Figure 11c-d**, some cells escape conditional Nanog deletion. At E7.5 Nanog null embryos are present at 50% of the expected Mendelian ratio and those that are present are delayed compared to wildtype littermates (**Supplementary Figure 12b**). It is well known that the embryo can suffer deletion of epiblast cells and thereafter recover (Snow and Tam 1979 DOI 10.1038/279555a0). Since Nanog is spatiotemporally limited, we hypothesise that if null cells survive past a critical gastrulation-stage period, they may be viable thereafter. While this raises interesting questions about the molecular basis of developmental plasticity, we would argue that this is beyond the scope of the present study, which is focussed on the molecular interactions of pluripotency TFs during early post-implantation development.*

Does the posterior proximal epiblast of Sox2^{cre}/Nanog^{f/f} embryos give rise to EpiSCs?

*The experiment as suggested is difficult to perform, due to the low recovery of embryos of the desired genotype. However, we addressed this point in vitro showing that posterior epiblast-like cells derived from EpiLCs do not give rise to EpiSCs unless Sox2 expression is induced (**Fig3d-f and Supplementary Fig7d-f**).*

Point-by-point rebuttal (reviewer text in black, our response in red)

We note that in our original rebuttal, we answered in detail every point made by both reviewers. The reviewer has not commented on the vast majority of these arguments.

The reviewer mentions that we have added data:

1) RNA-seq analysis of aEpiLC and pEpiLC (Figure 4), 2) Test SOX2 levels after ectopic anterior expression of Nanog (Figure 5a and b).

The reviewer has either ignored or missed the new data where we have carried out a timecourse of NANOG-SOX2 anticorrelation in vitro (Fig 4d-e), and our bioinformatic analysis (Fig S8) that, together with our new RNA-seq data and timecourse data, points to a network of Nanog-regulated genes (Fig 4f-h).

However, their data and arguments are not convincing enough to address the critical concerns raised by both Reviewers #1 and #2.

The major concerns remain: 1) novelty in their findings and 2) the unconvincing data supporting their conclusion that NANOG represses SOX2.

1) novelty in their findings

As the authors mentioned, the segregation of SOX2 and NANOG have been well reported. Although the authors claim that segregated NANOG and SOX2 expression has not previously been quantified in dual immunostaining through development, nor has the importance of this expression pattern been pursued, such data has already been published (e.g. Mulas C. et al., (2018) Development. PMID: 29915126).

We had considered Mulas et al. (2018) carefully. In this paper, although SOX2 and NANOG co- immunostaining was performed, SOX2 and NANOG channels were not described simultaneously per nucleus, and so do not add to the data of Hoffmann et al. (2013) in this respect. Staining was only quantified in E6.75 and E7.0 embryos. They therefore did not establish when the SOX2/NANOG anticorrelation arises during the crucial peri-implantation period when NANOG expression is re-initiated. Our data in Fig 1 and Supplementary Fig 1 show NANOG expression appears, and within a <12 hour window, individual cells accumulating high NANOG at the posterior epiblast show SOX2 depletion. Furthermore Mulas et al. analysed the early postimplantation role of OCT4, not SOX2 or NANOG. In our manuscript we rule out OCT4 as the critical factor deciding whether the posterior primitive streak is pluripotent (Figure S5). We mined the bioinformatic data of Mulas et al, and found that there was no overlap between the differentially expressed genes they found when OCT4 levels were changed, and our SOX2-correlated and NANOG-correlated genelists. See the genelists below:

For brevity, we considered this paper outside the scope of our manuscript but have now included in the discussion the non-overlapping differentially expressed geneset, highlighting the specificity of the SOX2-NANOG axis.

Moreover, the journal and the reviewer required novelty in biological findings, rather than in methodology. It was not clearly explained what novel biological findings they found using their approach.

We have indeed carefully quantified both SOX2 and NANOG but this paper is not a methodological one. The novel biological findings are the repression of SOX2 by NANOG and consequent loss of pluripotency in the posterior epiblast.

2) the unconvincing data supporting their conclusion that NANOG represses SOX2

While it's clearly showed that cells with high NANOG showed dramatically decreased SOX2 levels in pre-streak and early-streak embryos (Figure 1a, b, c), in new Figure 3h and 3i, NANOG overexpression only slightly repressed Sox2 expression (Approximately x 0.7 in Figure 3h, Approximately x 0.5 in 3i, with large error bars).

It would not be reasonable to expect complete disappearance of a transcript within one hour of repression. The effect is highly statistically significant (P=0.011). Below, we show the individual biological repeats. In each instance, Sox2 nascent RNA is depleted 1 hour after tamoxifen-mediated nuclear translocation of NANOG (+OHT versus -OHT; y-axis, Sox2 nascent transcript/TBP). This visualisation of the results presented in Figure 3i shows that variation in Sox2 nascent transcript mainly concerns the starting levels of transcript (-OHT) rather than the magnitude of repression by Nanog (+OHT)

Therefore, these data suggest that the reduction of SOX2 observed in vivo is not primarily caused by NANOG.

This suggests that the reviewer now believes our conclusion that NANOG suppresses SOX2- already a novel and surprising finding since NANOG and SOX2 cooperate to promote naive pluripotency. It's unclear what they mean by 'primarily'. We understand them to mean NANOG + something else represses SOX2. We discussed this point in the manuscript, but have now expanded the discussion on the primacy of NANOG to include the points that SOX-NANOG anticorrelation is preserved when OCT4 is absent, arguing that OCT4 is not essential for this negative regulation, and to expand a little upon a potential indirect regulation of SOX2 by OCT6, which our data argues against. The 50% reduction in nascent transcript indicating direct transcriptional repression may not by itself be large enough to suppress pluripotency, but the NANOG-mediated reduction of SOX2 protein that we then show, is. The reviewer dismisses the rest of the findings based on this incomplete suppression of Sox2 transcript. However it is clear that SOX2 repression is caused by NANOG- see below.

This is further supported by new Figure 5a and 5b, which shows no negative correlation between NANOG GFP signal intensity and SOX2 signal intensity.

We cannot understand this interpretation of Figure 5a and 5b (negative correlation in 5b included below for convenience), which, together with Supplementary Figure 9 showing all the additional embryo-by-embryo data, shows negative correlation between NANOG-GFP and SOX2 intensity.

We also have data in EpiLC showing ectopic NANOG expression suppresses SOX2 that we had included in the previous version but removed in an effort to simplify the narrative (see below).

However if it is felt that this data is helpful, we can reinstate it to show that the suppression happens both in vitro and in vivo on ectopic NANOG expression.

Dear Val, dear Ian,

Thank you again for the submission of your amended manuscript (EMBOJ-2025-120853) to The EMBO Journal, as well as for your patience with our response in light of the unusual protraction due to delayed expert input. We have carefully assessed your manuscript and the point-by-point response provided to the referee concerns that were raised during review at a different journal. In addition, and as mentioned before, we decided to involve an arbitrating advisor to evaluate the revised version of your work, with respect to technical robustness, conceptual advance and overall suitability of your work for publication in The EMBO Journal.

As you will see from his/her comment enclosed below, the advisor is broadly in favour of the work stating the interest and value of your results and s/he is supportive of publication at The EMBO Journal.

We are thus pleased to inform you that we can offer to swiftly move forward towards acceptance of this work at The EMBO Journal.

We now need you to take care of a number of minor issues related to formatting and data annotation, which I will share shortly in a separate message, together with additional changes and requests by our production team and instructions for Source Data provision.

Please submit a revised version of the manuscript using the link enclosed below, addressing these minor issues.

As you might have seen on our web page, every paper at the EMBO Journal now includes a 'Synopsis', displayed on the html and freely accessible to all readers. The synopsis includes a 'model' figure as well as 2-5 one-short-sentence bullet points that summarize the article. I would appreciate if you could provide this figure and the bullet points.

Thank you again for giving us the chance to consider your manuscript for The EMBO Journal, I look forward to hearing from you and receiving your final revised version of the manuscript.

Kind regards,

Daniel

EMBOJ-2025-120853, arbitrating expert's advice:

I think this is an excellent manuscript and there is much that is novel. I know of no other paper that has done such careful regional microdissection of mouse post implantation embryos for ex vivo culture and functional tests with stem cell derivation. These are incredibly challenging experiments to do and the data presented provide novel insights into developmental and stem cell biology. The insights into Nanog and Sox2 regulation in this context are also interesting and contribute novel understanding. I feel that they have sufficiently addressed the peer reviewer comments and I would strongly support publishing the work as is.

Dear Val, dear Ian,

Further to below, please find enclosed the mentioned additional remarks regarding required adjustments to formatting and data annotation for your study.

Please approach us any time should there be questions related.

We look forward to your final resubmission.

Best regards,
Daniel

>> Please add up to five keywords to your study.

>> Author Contributions: Remove the author contributions information from the manuscript text. Note that CRediT has replaced the traditional author contributions section as of now because it offers a systematic machine-readable author contributions

format that allows for more effective research assessment. and use the free text boxes beneath each contributing author's name to add specific details on the author's contribution.

More information is available in our guide to authors.
<https://www.embopress.org/page/journal/14602075/authorguide>

>> Adjust the title of the 'Competing Interests' section to 'Disclosure and Competing Interests Statement'.

>> Provide a completed Author Checklist.

>> Section order should be corrected as follows: title page with complete author information, abstract, keywords, introduction, results, discussion, methods, data availability section, acknowledgements, disclosure and competing interests statement, references, main figure legends, tables, expanded figure legends. The Methods should be added to the main manuscript text, after the Discussion.

>> The "additional information" sentence should be removed from the manuscript text.

>> Figures in separate files: main figures and suppl. figures should be uploaded as separate files; the legends of the suppl. figures should be added to the manuscript text after the main figure legends and under the heading "Expanded View Figure Legends". The suppl. figures should be renamed "Figure EV1" - EV12.

>> - Figure 5 is currently in landscape format, please correct to portrait orientation

>> References: please adjust reference format to EMBO Journal format, 10 authors et al, and place References after the Discussion, before figure legends.

>> Figure callouts: callouts for Fig 3G and Dataset EV1-3 should be added to the main text; the panels for the supplementary figures should all be called out.

>> Please add a Reagents and Tools table to the Methods section, as a separate file using the existing template in the Guide For Authors, listing key reagents, experimental models, software and relevant equipment.

>> Dataset EV legends: the datasets should be renamed "Dataset EV1" - EV3; all three need legends added to the corresponding files in a separate tab/worksheet.

>> Please provide source data for the study as to the separate request e-mail by my colleague Hannah Sonntag. Source data should be uploaded as one (zipped) file per figure.

>> Funding: please enter the all funding information into the list of funders in our online system.

>> Add a separate 'Statistical Analysis' section to the Methods part, detailing the algorithms and statistical tests applied.

>> Data availability section: please add a separate 'Data availability' section to the manuscript, stating: 'No large-scale data sets amenable to public repository deposition were generated in this study.'

>> Consider additional changes and comments from our production team as indicated below:

- Figure legends:

1. Please note that the exact p values are not provided in the legend of figure 3E
2. Please note that the box plots need to be defined in terms of minima, maxima, centre, bounds of box and whiskers, and percentile in the legends of figures 2A, G
3. Please note that $n=2$ in figure 2B
4. Please note that the error bars are not defined in the legends of figures 3E, F.
5. Please note that the measure of center for the error bars needs to be defined in the legend of figure 2B
6. Please note that scale bar and its definition are missing for figures 2C, E, J.
7. Please note that the white arrow is not defined in the legend of figure 2C. This needs to be rectified.

Dear Val, dear Ian,

Sending just another brief note to share input by my office colleague overseeing manuscript quality checks. Please consider his remarks below.

Please let us know if there are any questions related.

Best regards,
Daniel

Additional data and annotation issues to be addressed for the revised version of the manuscript:

>> Please provide all in portrait format.

>> Please provide source data for Supplemental Figure 5F.

>> Many cells are highly pixelated (eg Figure 3E, 4E). Please provide the cells image files in the original 16bit captured image (rather than converted to jpeg or pdf) to avoid integrity issues.

>> Please indicate redisplay of data in the figure legends for Supplemental Figure 1A (Figure 2A), Supplemental Figure 11B (Figure 5C), Supplemental Figure 2B (Supplemental Figure 1A) and Supplementary figure 8C and - Figure 2B ii (PMID: 35901961, <https://doi.org/10.1016/j.gpb.2022.06.006> Elsevier).

The authors addressed the remaining formatting issues.

Dear Val, dear Ian,

Thank you for submitting the revised version of your manuscript. I have now evaluated your amended manuscript and concluded that the remaining minor concerns have been sufficiently addressed.

I am thus pleased to inform you that your manuscript has been accepted for publication in the EMBO Journal.

On a different note, I would like to alert you that EMBO Press offers a format for a video-synopsis of work published with us, which essentially is a short, author-generated film explaining the core findings in hand drawings, and, as we believe, can be very useful to increase visibility of the work. Please see the following link for representative examples and their integration into the article web page:

<https://www.embopress.org/doi/full/10.15252/emj.2019103932>

Kind regards,

Daniel

Daniel Klimmeck, PhD
Senior Editor
The EMBO Journal
EMBO
Postfach 1022-40
Meyerohofstrasse 1
D-69117 Heidelberg
contact@embojournal.org
